# Seatbelts and raincoats, or banks and castles: Investigating the impact of vaccine metaphors

**Stephen J. Flusberg**[1]*, **Alison Mackey**[2], **Elena Semino**[3]

1 Department of Cognitive Science, Vassar College, Poughkeepsie, New York, United States of America,
2 Department of Linguistics, Georgetown University, Washington, DC, United States of America,
3 Department of Linguistics and English Language, Lancaster University, Lancaster, United Kingdom

* sflusberg@vassar.edu

**Data Availability Statement:** Data are publicly available on the Open Science Framework at the following link: https://osf.io/jg9st/.

## Abstract

While metaphors are frequently used to address misconceptions and hesitancy about vaccines, it is unclear how effective they are in health messaging. Using a between-subject, pre-test/posttest design, we investigated the impact of explanatory metaphors on people's attitudes toward vaccines. We recruited participants online in the US ($N = 301$) and asked them to provide feedback on a (fictional) health messaging campaign, which we organized around responses to five common questions about vaccines. All participants completed a 24-item measure of their attitudes towards vaccines before and after evaluating the responses to the five questions. We created three possible response passages for each vaccine question: two included extended explanatory metaphors, and one contained a literal response (i.e., no explanatory metaphors). Participants were randomly assigned to receive either all metaphors or all 'literal' responses. They rated each response on several dimensions and then described how they would answer the target question about vaccines if it were posed by a friend. Results showed participants in both conditions rated most messages as being similarly understandable, informative, and persuasive, with a few notable exceptions. Participants in both conditions also exhibited a similar small—but significant—increase in favorable attitudes towards vaccines from pre- to posttest. Notably, participants in the metaphor condition provided longer free-response answers to the question posed by a hypothetical friend, with different metaphors being reused to different extents and in different ways in their responses. Taken together, our findings suggest that: (a) Brief health messaging passages may have the potential to improve attitudes towards vaccines, (b) Metaphors neither enhance nor reduce this attitude effect, (c) Metaphors may be more helpful than literal language in facilitating further social communication about vaccines.

## Introduction

The novel coronavirus pandemic claimed nearly seven million lives worldwide in the first three years. This outcome could have been much worse if not for the development of safe and effective vaccines that greatly reduced the risk of dying from COVID-19. While governments across the globe worked to make vaccines widely accessible—and, in many contexts, mandated

**Funding:** This work was funded by UK Research and Innovation, grant numbers: ES/R008906/1 and ES/V000926/1 to ES. The funders had no role in study design, data collection and analysis, decision to publish, or preparation of the manuscript.

**Competing interests:** The authors have declared that no competing interests exist.

their use—these efforts were accompanied by a vocal backlash and an outpouring of anti-vaccination sentiment. This resistance was not a surprise for some observers. The World Health Organization (WHO) had named "vaccine hesitancy" one of the top threats to global health in 2019 [1], months before the pandemic would take hold.

Vaccine hesitancy is a complex phenomenon [2]. It has been associated with many factors, including age, education, mistrust in institutions, engaging with misleading sources online, local and sometimes vaccine-specific personal/family histories [3], and "folkloric narratives" [4–8]. A 2014 WHO report from the Strategic Advisory Group of Experts on Immunization (SAGE) includes three categories of determinants of vaccine hesitancy [9]: (a) "contextual influences" (e.g., religion, culture, politics, media environment); (b) "individual and group influences" (e.g., previous experiences with vaccinations by the individual and their kinship and social groups, immunization as a social norm or as not needed or harmful); and (c) "vaccine/vaccination-specific issues" (e.g., new vaccine, mode of administration, cost, risks vs. benefits). Responding to vaccine hesitancy is, therefore, likewise a complex enterprise that goes beyond the provision of 'accurate' information [10], whether in public health campaigns or interactions between healthcare providers and individuals. In this context, then, it is important to investigate the utility of different approaches to vaccine-related communications. The current study investigates the effectiveness of explanatory metaphors in public health messaging about vaccination and their influence on how individuals perceive and communicate about different aspects of vaccinations with members of their own social groups.

## Metaphor in communication and thought

Metaphor has been regarded as an important tool for persuasion and explanation since classical antiquity. Over the last 50 years, scholars from multiple disciplines have studied the role of metaphor in communication and thought using a broad range of methods. The seminal work by linguist George Lakoff and philosopher Mark Johnson has proven especially influential in this pursuit. They famously proposed that conventional metaphors in language (e.g., "She *shot down* all my arguments") are realizations of conceptual metaphors in thought (e.g., ARGUMENT IS WAR; [11]). In other words, people don't just *speak* metaphorically; they *think* metaphorically.

According to Lakoff and Johnson's 'Conceptual Metaphor Theory' (CMT), we understand a *target* domain (e.g., ARGUMENT) by importing knowledge and structure from a *source* domain (e.g., WAR). In this way, people can leverage their knowledge of familiar subjects to reason about novel or complex subjects. The choice of source domain influences how the target domain is conceptualized, resulting in potential "framing effects" of metaphors. This is because metaphors have unique *entailments*; the structure of the source domain facilitates certain inferences—but not others—when it is mapped onto the target domain. For example, if an argument is understood metaphorically as a *war*, then the goal of interactants is to *win*—to ensure that one's own views prevail. If an argument is understood metaphorically as a *journey*, in contrast, then the goal of interactants may be to *meet in the middle*—to find a compromise. Lakoff and Johnson argued that human thought is largely organized in this way and that reasoning about abstract domains is virtually impossible without metaphors [11].

Discourse analysts have, in some cases, found Lakoff and Johnson's notion of the conceptual domain too broad to explain how metaphors are used in discourse, especially in the context of explanation and persuasion. In a series of studies on political discourse, Musolff [12, 13] has employed the sub-domain notion of *scenario* as a "a set of assumptions made by competent members of a discourse community about 'typical' aspects of a source-situation, for example, its participants and their roles, the 'dramatic' storylines and outcomes, and conventional

evaluations of whether they count as successful or unsuccessful, normal or abnormal, permissible or illegitimate, etc." [12, p. 28]. In this way, the same overarching domain may generate different conceptual entailments under difference domain-related scenarios. For example, in a study of metaphors in cancer experience, Semino et al. [14] showed how *war*-related metaphors can have different implications (e.g., for the (dis)empowerment of patients) depending on the specific scenario involved, such as PREPARING FOR BATTLE (e.g., "I am *sharpening my weapons* in case I have to *do battle*") vs. OUTCOME OF BATTLE (e.g., "I'm not *winning* this *battle*").

While there are ongoing debates over CMT and the cognitive mechanisms that support metaphor processing, there is a general consensus that metaphors play a critical role in communication and cognition [15–18]. Over the past decade, for example, a large body of experimental research has shown that metaphors can enhance the persuasive power of messaging and explanations (for review, see [17, 18]). Using a metaphor to convey a complex issue like climate change [19], crime [20], cancer [21, 22], or the federal budget [23] can shape observer beliefs and attitudes in ways that are compatible with the metaphor's entailments.

The impact of framing a concept with a metaphor has been studied in a variety of ways. This includes having participants who have received information via metaphor provide or select a solution to a problem [20, 24], express their behavioral intentions [19, 21], and indicate their beliefs, preferences, or emotional states [22, 23, 53]. Table 1 provides a summary of variables that have been shown to moderate metaphor framing effects. This work has provided rich insights into the cognitive, affective, and social-pragmatic factors that influence the potency of metaphorical messages (see [17, 18] for reviews).

## Explanatory metaphors in vaccine discourse

Health communicators have developed a range of metaphors to address common questions and misconceptions about vaccines. These healthcare metaphors are often extended, elaborated, and used to provide explanations with the goal of persuasion. They contrast with the more subtle metaphoric language studied in most research on metaphor framing described above (although see [23] for an exception).

Recently, for instance, a range of metaphors have been deployed by scientists and healthcare practitioners to explain why people should get the COVID-19 vaccine even though it does not offer complete protection from the virus. For example, the US-based Northeast Georgia Health System published a blog post in 2021 titled *"How is the COVID-19 Vaccine like a Seatbelt?"* [33]. As the blog post reminds the reader, wearing a seatbelt keeps you safe in the event of an accident, but it is still possible to get hurt. This is also the case with vaccines for COVID-19. A similar idea is conveyed by the metaphor VACCINES ARE RAINCOATS, used, for example, in the following representative X post (formerly called Twitter) by a scientist in July 2021 [34]: "Concerning breakthrough infections. Think of the vaccine as a very effective raincoat. If it's drizzling, you'll be protected. If the rain is coming down hard, you might still be fine. But if you are going in and out of rainstorms all the time, you could end up getting wet."

Another common question about the COVID-19 vaccines is how they were developed so quickly—and whether this means they are not as safe as vaccines that were developed more slowly in the past. Professor Sarah Gilbert and Dr. Catherine Green used a vivid metaphor to address this concern in their book *Vaxxers* about the development of the Oxford AstraZeneca vaccine [35]. They ask the reader to consider how a baker might speed up the process of producing cakes with personalized messages (p. 66):

"Think about a baker who sells personalised cakes iced with a message like 'Happy 50th birthday Joe' or 'Congratulations on your engagement Ali and Max'. She might wait until

she gets an order, and only then start the process of mixing the ingredients, baking the cake, letting it cool, icing it all over, waiting for the icing to set, and then finally adding the customized message. If she gets the order the day before the cake is needed, that works well. But if she wants to be able to offer a quicker service, she could bake a stock of cakes and put on the base layer of icing every morning. She is taking a financial risk: if no orders come in, the pre-baked cakes will go stale and need to be thrown away. But it may be worth the risk. When a customer comes into the shop, all she has to do is pick up her piping bag and add the custom message while he waits. The cake is then ready to take straight to the party. Only in the case of a vaccine, the party is a pandemic."

In this metaphor, the cake represents the starting material needed to develop any generic vaccine. The personalized icing represents the pathogen-specific string of DNA that is added to create a pathogen-specific vaccine. As the authors explain, researchers working on COVID-19 vaccines had *pre-baked* a stock of *cakes* that could be quickly *iced* when the novel coronavirus was decoded. *Baking each personalized cake from scratch* when the order is placed, on the other hand, reflects the older, less efficient method for developing vaccines.

The authors use another food-based metaphor to supplement their explanation. Before the COVID-19 pandemic led governments worldwide to invest enormous sums into vaccine research, funding this work was a slow and arduous process that required a lot of bureaucratic red tape (p. 157): "It is as if you are making a roast dinner and for every ingredient you have to make a separate trip to the shops to buy it, then cook it and demonstrate that it is going to be delicious, before moving on to the next." When COVID-19 hit and governments infused this sector with substantial funding, on the other hand, "We were allowed to do a big shop and put all the ingredients we needed in the trolley all at once" (p. 158).

There has been limited research on the impact of metaphor framing on vaccine attitudes [36, 37]. In one such study, Scherer and colleagues [36] investigated whether describing influenza metaphorically (as a *beast*, *riot*, *army*, or *weed*) would increase people's willingness and interest in being vaccinated. They found that, compared to a comparable literal description, metaphorical descriptions of the flu increased people's expressed willingness to be vaccinated. This effect was observed more consistently in people who occasionally got the flu vaccine compared with those who never got it. Their study highlights one variable—the strength of prior beliefs or attitudes—that moderated the impact of a metaphoric message (as shown in Table 1, for example). Describing the influenza virus as a *beast* is quite different from the elaborate explanatory metaphors used in COVID-19 discourse, however. For one thing, *beast* is a metaphor for a virus rather than a metaphor for some aspect of vaccination. For another, *beast* is a subtle, one-off metaphor rather than an extended and elaborated explanatory metaphor. In contrast, the *cake* metaphor described above is explicitly presented as an explanation for how researchers could develop the COVID-19 vaccines so quickly, and it was extended and developed throughout the text. It is this latter type of *explanatory* metaphor that we aimed to evaluate in the present study.

A recent study by Ervas and colleagues [37] did examine how an explanatory metaphor would impact the efficacy of vaccine communications. Their central message focused on "collective immunity," emphasizing how all people should be required to receive a vaccination to benefit the larger group. Including an explanatory metaphor—likening people to *bees in a beehive* who must collaborate—significantly increased a range of assessments of the message, including its emotional impact and perceived convincingness. However, it did not increase intentions to vaccinate. Our study builds on this basic design by sampling a wider range of more elaborate explanatory metaphors in vaccine discourse and using a more comprehensive measure of general vaccine attitudes.

**Table 1. Variables that moderate the impact of metaphor framing.**

| Variable | Findings | References |
|---|---|---|
| Source domains | Metaphors are more likely to have an influence if they involve source domains that are accessible, well-delineated, and image-rich. | [17, 18, 53] |
| Target domains | Metaphors are more likely to have an influence if they involve target domains that are neither too unfamiliar/uninteresting for people nor overly linked to strong prior beliefs and evaluations. | [17, 18] |
| Structural alignment between source and target domain | Metaphors are more likely to have an influence if they involve precise and clearly applicable mappings, i.e., when they are 'apt'. | [17, 23] |
| Textual position | Metaphors are more likely to have an influence if they are presented early rather than late in a text. | [20] |
| Extension | Metaphors are more effective when they are extended. | [25–27] |
| Conventionality/creativity | Metaphors are more likely to be appreciated and found meaningful if they are conventional or moderately innovative as opposed to highly innovative, likely because radically novel metaphors can be difficult to understand and harder to appreciate. | [28, 29] |
| Participant characteristics | The influence of metaphors varies depending on how much the source domain resonates with the individual, their political affiliation, and cultural background; for example, strong prior beliefs can mitigate the impact of a metaphorical message. | [24, 30–32] |

## The current study

A critical question concerns the general effectiveness of extended explanatory metaphors in vaccine discourse. That is, do such metaphors lead to positive effects on people's understanding of and attitudes towards vaccination? To date, this issue has not been systematically assessed in the experimental literature. We addressed this question in the current study by comparing the outcomes of participants of health messages with and without extended explanatory metaphors.

## Methods

### Overview

We used a between-subject, pretest/posttest design. Participants completed a 24-item measure of their attitudes towards vaccines both before and after evaluating health messages that we organized as a series of responses to five common questions about vaccines. We created three responses for each question: two responses included unique explanatory metaphors—adapted from real-world source materials like the ones described above—and a baseline response that provided the same information but did not include an explanatory metaphor. We refer to these baseline responses as 'literal' responses, and we operationalized this term as "not containing an extended explanatory metaphor." Participants were randomly assigned to receive either all metaphor or all literal messages. Those in the metaphor condition randomly received one of the two explanatory metaphors in response to each of the five target questions. After reading each response, metaphor and literal participants rated how understandable, informative, and persuasive it was. This allowed us to measure whether people interpreted metaphor-infused messages as more effective, equally effective, or less effective when compared to literal messages (i.e., those that were not organized around a central explanatory metaphor). We also included an innovative method for assessing the impact of metaphorically framed messages: after rating each message, participants were asked to freely describe how they would answer the associated target question if it were posed to them by a friend. This resulted in a rich

natural language dataset that allowed us to conduct novel linguistic analyses to assess the impact of metaphors.

Our findings provide important insights into the effects of explanatory metaphors in vaccine communications. The study also serves as a design model for the systematic investigation of the role of explanatory metaphors in public discourse, as we included a combination of traditional quantitative and linguistic analytic methods.

## Participants

Data were collected in June 2022. We recruited 301 participants from Amazon's Mechanical Turk crowdsourcing platform [38]. We used the CloudResearch interface [39], which includes a set of pre-screening measures shown to improve Mturk participant and data quality [40, 41]. We aimed for a sample size of at least 100 individuals in each condition to be consistent with past research on metaphor framing [e.g., 17, 25, 42, 43]. All participants were at least 18 years of age ($M = 38.3$, $SD = 10.4$), residents of the United States, and had a good performance record on previous MTurk tasks (with a minimum of 95% approval rating). The sample included 117 female, 176 male, and four non-binary/no-gender participants, with four additional participants who preferred not to indicate their gender. We did not collect information that could identify individual participants during or after data collection. All participants gave informed consent prior to beginning the study. This study was reviewed and approved under reference FASSLUMS-2021-0576-RECR-2 by Lancaster University Management School Research Ethics Committee's Faculty of Arts and Social Sciences (FASS-LUMS). Participants were paid $3 (USD) once they completed the study.

## Materials

**Vaccine Attitude Measure (VAM).**    We developed a 24-item questionnaire designed to measure participants' beliefs and attitudes toward vaccines. Most items were adapted from (a) the Vaccination Attitudes Examination Scale [44], (b) the Vaccine Hesitancy Scale [45], and (c) the Vaccine Conspiracy Beliefs Scale [46]. We also included (d) unique items generated for the purposes of the current study. We drew on multiple sources to create a comprehensive measure that addressed a wide range of vaccine beliefs and attitudes. Items were clustered into eight sub-categories consisting of three items each, as illustrated in Table 2. The instructions required participants to indicate their support for a series of statements on a scale from 1 (*strongly disagree*) to 7 (*strongly agree*). Tests of internal consistency showed the measure was reliable (Cronbach's $\alpha = 0.937$ for the VAM pretest). In coding, we averaged together responses, reverse-coding as needed, such that higher scores equate to more positive attitudes and more accurate beliefs about vaccines.

**Vaccination questions and messages.**    We designed the stimulus materials to resemble a public health messaging campaign. To identify common misperceptions, we focused on public health messaging in the U.S., the U.K., and other international organizations. This included, for example, the comprehensive *Myths and Facts about COVID-19 Vaccines* from the Centers for Diseases Control in the U.S. [47], *Vaccine Myths* from the British Society for Immunology [48], and *Vaccines and immunization*: *Myths and Misconceptions* from the World Health Organization [49]. We focused on five questions aimed at tapping into common questions, misconceptions, and concerns about vaccines, as follows:

1. How do vaccines work?

2. Is 'natural immunity' better than the immunity provided by a vaccine?

3. Are vaccines that are developed quickly safe?

**Table 2. Items, coding, and sub-categories in our Vaccine Attitude Measure (VAM).**

| Item | Sub-Category |
| --- | --- |
| Vaccines are effective at preventing serious illness | How vaccines work |
| Vaccines are designed to target and neutralize viruses that enter the body (I)[a] | |
| Vaccines contain new antibodies designed to deal with infections (I)[a] | |
| Natural immunity is better than immunity achieved through vaccination (I)[a] | Natural immunity |
| Natural exposure to viruses and germs gives the most effective protection (I)[a] | |
| Being exposed to diseases naturally is safer for the immune system than being exposed through vaccination (I)[a] | |
| Each new disease requires that a new vaccine be made from scratch (I)[a] | Speed of vaccine development |
| Vaccines that are developed quickly cannot be trusted (I)[a] | |
| Newer methods allow safe vaccines to be developed more rapidly than in the past | |
| Only people who are at risk of serious illness should get a vaccine (I)[a] | Personal versus community risk |
| Vaccines only impact the individual who gets the vaccine (I)[a] | |
| Getting vaccinated is important for the health of others in my community | |
| Vaccines that do not fully prevent infections are ineffective (I)[a] | Why vaccinate if not 100% effective |
| If a vaccine requires multiple doses or boosters, it means the vaccine isn't effective (I)[a] | |
| Vaccines prevent you from spreading a virus to other people (I)[a] | |
| I do not have concerns about getting vaccinated | Personal attitudes towards vaccines |
| I can rely on vaccines to stop me from getting seriously ill from an infectious disease | |
| I feel protected after getting vaccinated | |
| Immunizing children is harmful, and this fact is covered up (I)[a] | Attitudes toward childhood vaccinations |
| Childhood vaccines are important for public health | |
| Getting vaccines is a good way to protect children from disease | |
| The government is trying to cover up the link between vaccines and autism (I)[a] | Vaccine conspiracy beliefs |
| Pharmaceutical companies cover up the dangers of vaccines (I)[a] | |
| Vaccine efficacy data is often fabricated (I)[a] | |

[a] "I" indicates a reverse-coded item.

4. Why should I take a vaccine if I am personally at low risk for the illness?

5. Why get a vaccine if it isn't 100% effective?

For each question, we created three corresponding responses: two that used extended explanatory metaphors that were different from each other and a baseline literal response that, as we noted earlier, did not include an extended, explanatory metaphor. Using Musolff's [12] terms, these extended metaphors exploit the narrative potential of specific source scenarios. Most of the metaphors were adapted from authentic, real-world examples of metaphors in common use in vaccine discourse. These were collected through the authors' observations from the media and from websites on public health messaging, like those listed above and described in the Introduction. We computed readability consensus scores to ensure comparability of the texts based primarily on the work of Flesch and Flesch-Kincaid and others [50], using a range of measures to gauge the readability or difficulty level of a text in English. These explanatory messages are provided in Table 3.

After reading one of the three explanations associated with a particular question, participants were asked to consider how the general public would react to the message and to keep this in mind while responding to four questions. Participants were asked to consider the general public in order to reduce any reluctance they might have to go on the record with their

**Table 3. Vaccination questions and corresponding explanation stimuli.**

| | |
|---|---|
| *Question 1*: *How do vaccines work*? | |
| Metaphor 1: *Castle* | To understand the reason for vaccinations, it's important to understand how vaccines work. Vaccines enable your immune system to do two things: (1) stop you getting infected by viruses or, (2) if you do get infected, end the infection itself because of the help your immune system has had from the vaccine. You can picture your body as a medieval castle. The castle is surrounded by an army of viruses trying to break in and take over. Your body's first line of defense is an outer wall patrolled by a group of archers. These are your immune system's antibodies. If they can hold the viral army off, then you won't get infected. But if the antibody archers are overwhelmed, then the virus can break through. Once the virus is in the castle, you have an infection. However, all is not lost. You still have elite troops inside the castle. These are your memory B and memory T cells. If your outer walls are breached, these elite cells are inside ready to lead the charge and repel the hostile invaders. Vaccines train your body's troops, including both the archer-antibodies and the inner cell warriors that react to an infection. However, the antibodies that patrol the outer wall sometimes forget their training. That is when the vaccine's effectiveness wanes, and you may be infected even if you had previously been vaccinated. But the memory cells inside the inner castle are still there and can get organized very quickly to repel any invaders who have entered the castle. That is when a vaccine protects you from serious illness and death, even if not from infection. For viruses that won't go away or that even change over time, boosters can strengthen your defending army. Boosters not only ensure there are enough archers in position defending the outer wall, they also provide the elite troops inside the inner castle with updated weapons training so that they are prepared to take on any potential invaders. The archers and troops might still be able to do their job without boosters. However, since boosters typically contain updated information about the virus' new strengths and weaknesses, the archers and troops who face the infection without boosters are disadvantaged, with fewer resources and less knowledge than those who received the vaccine booster. |
| Metaphor 2: *Bank* | To understand the reason for vaccinations, it's important to understand how vaccines work. Vaccines enable your immune system to do two things: (1) stop you getting infected by viruses or, (2) if you do get infected, end the infection itself because of the help your immune system has had from the vaccine. You can picture your immune system as the high-quality security measures surrounding a bank. The first line of protection from burglars is the security cameras monitoring the doors and windows. These are your immune system's antibodies. If they detect the burglars and trigger the alarm, then you won't get infected. But if any of the cameras fails, then the virus can get in. You now have an infection. Any cash stored in the bank could quickly be stolen. However, all is not lost. You still have a team of security guards inside the bank. These are your memory B and memory T cells. If the bank is broken into, they are ready to lead the immunological charge to stop the burglars. Vaccines train your body's security-measures, including both the security camera antibodies and the inner cell guards that react to an infection. However, cameras can fail if they have not been updated for some time. That is when the vaccine's effectiveness wanes, and you may be infected even if you had previously been vaccinated. But the team of security guards is still there to deter an infection and can get organized very quickly to chase the burglars away before they steal the valuables in the bank. That is like when a vaccine protects you from serious illness and death even though it doesn't protect you from the initial infection. For viruses that won't go away or that even change over time, boosters can upgrade the bank's security measures. Boosters not only ensure the camera systems are up-to-date, but also provide the security guards inside the bank with new training and equipment so that they are prepared to repel any burglars. The camera systems providing outer-defense and the guards providing inner-defense may technically still be able to complete their responsibilities without boosters. However, since boosters typically contain updated Information about the virus' new strengths and weaknesses, the security systems and trained guards who have to face the infection without boosters are at a disadvantage with fewer resources and less knowledge than those who received the vaccine booster. |
| Literal | To understand the reason for vaccinations, it's important to understand how vaccines work. Vaccines enable your immune system to do two things: (1) stop you getting infected by viruses or, (2) if you do get infected, end the infection itself because of the help your immune system has had from the vaccine. One important factor in stopping infections is the presence of antibodies. If you have them in sufficient numbers, then you won't get infected. But if you don't have enough antibodies, the virus can start replicating in your cells and you get an infection. However, all is not lost. There are still your memory B and memory T cells. They can eliminate the infection in your body so that you recover without getting very sick. Vaccines enable your body to develop both the antibodies that help prevent infection and also the memory cells that can react to a potential infection. However, the antibodies decrease in number over time. As they decrease, the vaccine's effectiveness wanes, and you may become infected even though you've been vaccinated. However, your memory B and T cells last much longer, and are always ready to deal with an infection. This is how a vaccine prevents serious illness and death even when it doesn't prevent infection. For viruses that won't go away or that even change over time, boosters can both increase the number of antibodies again as well as strengthen the memory cells in case they are needed. |
| *Question 2*: *Is "natural immunity" better than the immunity provided by a vaccine*? | |
| Metaphor 1: *Pilot* | Vaccines help your body build up immunity safely without the risks associated with a viral infection. In this way, vaccines are like the flight simulation programs that airplane pilots are trained on before they attempt to fly a plane through bad weather conditions. This training allows pilots to practice flying through (simulated) bad weather in a safe environment, helping them become better pilots in real life later on. In the same way, vaccines help to train your immune system to handle a virus without the risks associated with a real infection. When an untrained pilot encounters bad weather for the first time while actually flying, they may be able to figure out how to get through it without crashing the plane, and having that experience would help them fly in bad weather better in the future. However, there's always more risk of crashing when pilots have no prior training. Similarly, there is always more risk that a person can become very ill from an infection if they have no prior immunity. In other words, immunity can be achieved either by being infected with a virus or by being vaccinated against it. However, with vaccines there is little risk of serious illness in this process. |

*(Continued)*

**Table 3.** (Continued)

| Metaphor 2: *Fire Drill* | Vaccines help your body build up immunity safely without the risks associated with a viral infection. In this way, vaccines are like the fire drills that students do in schools. These drills allow them to practice what they would do in a fire emergency but in a safe environment, so they would be better able to handle a real fire if one occurred. In the same way, vaccines help train your immune system to handle a virus without the risks associated with a real infection. If untrained students experience a fire in the classroom, they may be able to figure out how to get through it without getting injured, and this experience would help them survive a similar fire in the future. However, there's always more risk of panicking and getting seriously hurt in fire situations without training. Similarly, there is always more risk that a person can become very ill from an infection if they have no prior immunity. In other words, immunity can be achieved either by being infected with a virus or by being vaccinated against it. However, with vaccines there is little risk of serious illness in this process. |
|---|---|
| Literal | Vaccines help your body build up immunity safely without the risks associated with a viral infection. If an unvaccinated person recovers from a viral infection, their immune system may be able to protect them from similar viruses in the future. However, there is always a risk that a person might become very ill from the actual infection because they have no prior immunity. In other words, immunity can be achieved either by being infected with a virus or by being vaccinated against it. However, with vaccines there is little risk of serious illness in this process. |

*Question 3*: *Are vaccines that are developed quickly safe*?

| Metaphor 1: *Cake* | Scientists are now able to develop vaccines much more quickly than in the past. Why is this? Think about a baker who sells personalized cakes with messages like 'Happy 50th birthday, Taylor!' or 'Congratulations on your new job, Sam!'. The baker might wait until they receive an order, and only then start the baking process from scratch, mixing the necessary ingredients, baking the cake, letting it cool, making the icing, and finally adding the customer's personalized message. But this is a slow process. For a much more efficient service, bakers can make a batch of cakes ahead of time, and when a customer comes into the shop, they can ice the personalized message onto the pre-made cake. In a shorter amount of time, the personalized cake Is ready. In the same way, today's scientists do not start inventing vaccines from scratch when a new virus comes along. After many years of research and testing they have developed a vaccine-base, like a generic cake. When a new virus emerges, they quickly get the information they need to adapt the vaccine to the virus, which Is like adding messages on the cake. This new vaccine is immediately ready to be tested before being rolled out to the public. |
|---|---|
| Metaphor 2: *Video Game* | Scientists are now able to develop vaccines much more quickly than in the past. Why is this? Think about the companies that produce video games. Rather than creating a brand-new video game each time players ask for new features like storylines, characters, or gameplay modes, companies can meet their players' wishes by releasing updates to pre-existing games that can be downloaded to players' existing consoles. The developers simply write a piece of new code. They can then release game updates or patches that add these new features to their pre-existing video games, allowing players to continue playing their game with the latest features, all without needing to develop a brand-new game from scratch. These updates and patches with new game features are alterations to the original base program that can easily be tested and refined before finally being released to the public. In the same way, today's scientists do not start inventing vaccines from scratch when a new virus comes along. They have already developed an existing vaccine-base after many years of research and testing. When a new virus emerges, they quickly get the information they need to adapt the vaccine to the virus, which is like updating their base with the necessary and latest features. This new vaccine is immediately ready to be tested before being rolled out to the public. |
| Literal | Scientists are now able to develop vaccines much more quickly than in the past. Why is this? Nowadays, scientists do not start inventing vaccines from scratch when a new virus comes along. After many years of research and testing, they have developed generic technologies and platforms that can be used and adapted to fit any virus. When a new virus emerges, scientists quickly get the information they need to adapt the generic platform and create a new vaccine that is immediately ready to be tested before being rolled out to the public. |

*Question 4*: *Why should I take a vaccine if I am personally low-risk for the illness*?

| Metaphor 1: *Speed Limit* | With certain viruses, some people who get infected experience only minor symptoms while others can get very sick or even die. When there is uncertainty with new viruses, however, everyone is invited to get vaccinated, regardless of their own personal risk level. Likewise, speed limits apply equally to all drivers and vehicles. Even though different vehicles have different safety features for occupants, it doesn't make sense to allow some people to drive faster based on the safety features of their personal vehicle. This is because, even though some drivers might at lower risk of personal injury because of their vehicle's safety features, those drivers would still be a danger to other drivers on the road. In the same way, unvaccinated low-risk people may be less likely to be harmed by a virus, but they are still a danger to people who are more likely to be made very ill by the virus. |
|---|---|
| Metaphor 2: *War* | With certain viruses, some people who get infected can experience minor symptoms while others can get very sick or even die. When there is uncertainty with new viruses, however, everyone is invited to get vaccinated, regardless of their own personal risk level. Likewise, when a country is attacked in war, its leaders mobilize people, weapons, and resources from throughout the different regions of the country to defend it. This is because, even though some regions are low-risk, meaning less vulnerable to attack than others, it wouldn't make sense to exempt them from contributing, because cooperation across regions makes it more likely the country can repel the invaders. In the same way, unvaccinated low-risk people may be less likely to be harmed by a virus, but they can support the health of entire population by getting vaccinated, which helps the nation fight off the virus and reduces the chances of high-risk people becoming ill. |
| Literal | With certain viruses, some people who get infected experience minor symptoms while others can get very sick or even die. When there is uncertainty with new viruses, however, everyone is invited to get vaccinated, regardless of their own personal risk level. This is because, if low-risk people are unvaccinated, they may still infect other people who are more likely to be made very ill by the virus. So, unvaccinated people can be a danger to others, even if the chance of them becoming very ill is small. |

*Question 5*: *Why get a vaccine if it isn't 100% effective*?

(*Continued*)

**Table 3.** (Continued)

| | |
|---|---|
| Metaphor 1: *Raincoat* | Being vaccinated is an effective way of reducing your chances of being infected with a virus, just like wearing a waterproof raincoat during a storm reduces your chances of becoming wet. However, even the best raincoats don't provide 100% protection from getting wet. In the same way, vaccines don't provide 100% protection from a virus, and it is still possible to become sick even after you have been vaccinated and given a booster. For example, you could get sick if you are exposed to a large amount of the virus in your daily life, which is like going out during a severe rainstorm. You could also get sick if the immune reaction caused by the vaccine has not been strong or if a new variant develops that partly evades the vaccine. This is like a raincoat not fitting you well. Also, the effects of the vaccine might wane eventually, just like a raincoat might fray, develop holes, and wear out over time. For these reasons, when there are high infection rates in your area, it is still important to take additional precautions even after vaccination. These include avoiding crowded indoor spaces (as you would avoid severe rainstorms) and wearing a face mask (like using an umbrella even though you are also wearing a raincoat). |
| Metaphor 2: *Seatbelt* | Being vaccinated is an effective way of reducing your chances of being infected with a virus, just like wearing a seatbelt reduces your chances of getting injured in a car crash. However, even the best seatbelts don't provide 100% protection from getting hurt. In the same way, vaccines don't provide 100% protection from the virus, and it is still possible to become sick even after you have been vaccinated and given a booster. For example, you could get sick if you are exposed to a large amount of the virus in your daily life, which is like spending a lot of time in heavy, fast traffic. You could also get sick if the immune reaction caused by the vaccine has not been strong or if a new variant develops that partly evades the vaccine. This is like a seatbelt not fitting you well. Finally, the effects of the vaccine might wane eventually, just like a seatbelt might become less effective due to age and wear and tear. For these reasons, when there are high infection rates in your area, it is still important to take additional precautions even after vaccination. These include avoiding crowded indoor spaces (as you would avoid reckless driving) and wearing a face mask (like driving a car with airbags even though you are also wearing a seatbelt). |
| Literal | Being vaccinated is an effective way of reducing your chances of being infected with a virus. However, vaccines do not provide 100% protection, and it is still possible to become sick even after you have been vaccinated and had a booster. For example, you could get infected if were exposed to a large amount of a virus in your daily life. You could also get infected if the immune reaction caused by the vaccine has not been strong, or if a new variant develops that partly evades the vaccine. Finally, the effects of the vaccine might wane eventually. For these reasons, when there are high infection rates in your area, it is still important to take additional precautions even after vaccination, like avoiding crowded indoor spaces and wearing a face mask. |

personal responses given the highly emotive and politicized nature of vaccine/booster discourse at the time of the study, as well as to reduce any potential audience effect. In other words, the goal was to obtain maximally authentic data. First, participants rated three properties of the message using a 7-point scale: (1) "How understandable was this paragraph?" (1 = *very difficult to understand*, 7 = *very easy to understand*); (2) "How informative was this paragraph?" (1 = *not informative at all*, 7 = *very informative*), and (3) "How persuasive was this paragraph?" (1 = *not persuasive at all*, 7 = *very persuasive*). Finally—advancing a new methodology for metaphor research and mindful of the influence of social networks on vaccine attitudes—we asked participants how they would respond to a friend who asked them one of the five common questions above about vaccines (e.g., Question 5: "Why get a vaccine if it isn't 100% effective?"), and to provide the response they would give their friend by typing into a blank text box.

**Additional measures.** Participants completed two additional measures that pilot testing had revealed to be predictive of attitudes towards vaccines. These measures were, first, the Generic Conspiracist Beliefs (GCB) Scale, which is a 15-item measure of belief in generic conspiracy theories [51]. Examples include "The government is involved in the murder of innocent people and/or well-known public figures and keeps this a secret" and "Technology with mind-control capacities is used on people without their knowledge." We fine-tuned some of the language in the statements to enhance readability and ensure that the measure could be better understood and applied to participants outside the U.S. in the future. Participants rated the degree to which they believed each item was true on a scale from 1 (*definitely not true*) to 5 (*definitely true*). The measure had reliable internal consistency (Cronbach's $\alpha = 0.956$) and was coded by averaging all responses such that higher scores equate to a stronger belief in generic conspiracies.

We also used a second scale, the Trust in Institutions Scale (TIS), which was adapted from recommendations provided by the Organization for Economic Cooperation and Development [52]. On this scale, participants rate how much they personally trust the government, the

media, the education system, public health officials, political parties, and scientists on a scale from 0 (*do not trust at all*) to 6 (*complete trust*). The measure had reliable internal consistency (Cronbach's α = 0.905) and was coded by averaging all responses such that higher scores equate to greater trust in institutions.

**Demographics questionnaire.** Participants were asked to provide their age, gender, race/ethnicity, highest level of education achieved, and approximate household income, all using fill-in-the-blank text boxes. Participants were invited to leave the text box blank if they preferred not to provide a response. Participants were also asked to self-report their mathematics and science background using a 4-point scale (1 = *no background*, 2 = *not much background*, 3 = *some background*, 4 = *a lot of background*, following [48]). A few participants selected two adjacent points on the scale (e.g., 3 = *some background* and 4 = *a lot of background*), which led us to score them using the mean of the two points (i.e., 3.5 in the example). Participants also provided their political beliefs using a 7-point scale from 1 (*very liberal*) to 7 (*very conservative*). Finally, participants were asked to provide their COVID-19 vaccination status ("Have you had, or do you plan to get, a COVID-19 vaccination?" with response options *Yes*, *No*, and *Prefer not to say*), COVID-19 booster status, ("Have you had, or do you plan to get, a COVID-19 booster vaccination?" with response options *Yes, No*, and *Prefer not to say*), and their attitude towards vaccines in general (7-point scale from 1 = *strongly opposed to vaccines* to 7 = *strongly in favor of vaccines*).

## Procedure

**Attention check.** Participants first completed an attention check measure embedded in a paragraph ostensibly about how they would prefer to receive information about the study ("[...] Thus, in order to demonstrate that you are a participant who reads the study instructions carefully and thoroughly, you need to check the option "Other" below and enter the number 8 in the text box for this option"). Participants who answered incorrectly were thanked for accessing the survey but were unable to proceed with the study. Participants who answered correctly were directed to complete an informed consent form and proceed with the study.

**Instructions.** First, participants were told we were "interested in how people understand and react to public health messages" and that they would "read a series of excerpts from a non-partisan public health campaign about vaccines" and provide feedback about each message. They were also asked to complete the survey in one sitting without any breaks. Next, they completed the Vaccine Attitude Measure (VAM pretest). The order of the statements was randomized. This was followed by the vaccine questions and message stimuli. Participants received all five vaccine questions with their corresponding response questions in order from one to five, presented sequentially on separate screens. One-third of participants were randomly assigned to receive the literal messages for all five questions. The other two-thirds of participants were randomly assigned to receive one of the two metaphorical messages for all five questions. The specific metaphorical message they received for each question was randomly drawn from the two options. Next, participants completed the two additional measures (GCB scale and TIS) and the demographics questionnaire. Finally, they were debriefed, thanked for their time, and provided with the contact information for one of the authors. Participants took approximately 19 minutes on average to complete the entire survey (*SD* = 8.6 minutes).

## Results and discussion

### Participant vaccination data

The average level of support for vaccines, in general, was relatively high among our participants (*M* = 5.93, *SD* = 1.46 on our 7-point scale). Most participants reported they had received

or planned to receive a COVID-19 vaccination (81.4%) and booster (64.8%). The vaccination rate is in line with the overall U.S. rate of vaccination rate of 78% as of June 29, 2022 (the day after data collection was completed. See https://usafacts.org/visualizations/covid-vaccine-tracker-states). Only 32% of Americans had received a booster as of June 29, 2022, suggesting our sample was more likely to be boosted than the general public. However, our question about booster status also left open the possibility that a participant was planning to receive a booster shot in the future, indicating this gap is likely smaller than it appears. Overall, these data suggest our sample was slightly more accepting of COVID-19 vaccines than the general public.

## Explicit ratings of the health messages

All analyses were carried out using jamovi open-source statistics software. We conducted a series of one-way ANOVAs to compare the effect of each of the three stimulus vignettes for each of our five questions on ratings of understandability, informativeness, and persuasiveness. These analyses revealed that participants generally rated the metaphorical and literal responses for each question as similarly understandable, informative, and persuasive. There were, however, several notable significant differences, as revealed by Tukey's post-hoc $t$-tests: For Question 1, the *bank* ($M = 5.86$, $SD = 0.95$) and *castle* ($M = 5.89$, $SD = 1.42$) metaphors were rated as more understandable than the literal message ($M = 5.35$, $SD = 1.19$); $t(298) = 2.96$, $p = 0.009$ and $t(298) = 3.18$, $p = 0.005$, respectively. For Question 4, the *war* metaphor was rated as both less understandable ($M = 5.53$, $SD = 1.27$ vs. $M = 5.99$, $SD = 1.13$; $t(298) = 2.66$, $p = 0.023$) and less informative ($M = 5.00$, $SD = 1.56$ vs. $M = 5.54$, $SD = 1.27$; and $t(298) = 2.63$, $p = 0.025$) than the *speed limit* metaphor. The *war* metaphor was also rated as less understandable than the literal message ($M = 6.07$, $SD = 1.25$; $t(298) = 3.13$, $p = 0.005$). Finally, for Question 5, the *raincoat* metaphor was rated as more persuasive than the literal message ($M = 5.24$, $SD = 1.69$ vs. $M = 4.35$, $SD = 1.93$; $t(298) = 3.60$, $p = 0.001$). See Table 4.

**Table 4. Explicit ratings of each metaphor.**

| Question | Stimulus | Mean Rating (SD) | | |
|---|---|---|---|---|
| | | *Understandable* | *Informative* | *Persuasive* |
| 1. How do vaccines work? | *Castle* | 5.89 (1.42) | 5.82 (1.44) | 5.29 (1.60) |
| | *Bank* | 5.86 (0.95) | 5.71 (1.19) | 5.35 (1.29) |
| | Literal | 5.35[aa] (1.19) | 5.98 (1.07) | 5.17 (1.39) |
| 2: Is "natural immunity" better than the immunity provided by a vaccine? | *Pilot* | 5.80 (1.21) | 5.54 (1.46) | 5.16 (1.62) |
| | *Fire Drill* | 5.96 (1.04) | 5.50 (1.35) | 5.22 (1.50) |
| | Literal | 6.04 (1.32) | 5.36 (1.33) | 5.26 (1.62) |
| 3: Are vaccines that are developed quickly safe? | *Cake* | 5.97 (1.12) | 5.78 (1.23) | 5.42 (1.44) |
| | *Video Game* | 5.94 (1.14) | 5.58 (1.44) | 5.13 (1.66) |
| | Literal | 5.88 (1.16) | 5.46 (1.36) | 5.09 (1.60) |
| 4: Why should I take a vaccine if I am personally low-risk for the illness? | *Speed Limit* | 5.99 (1.13) | 5.54[b] (1.27) | 5.17 (1.73) |
| | *War* | 5.53[aa] (1.27) | 5.00[b] (1.56) | 4.78 (1.82) |
| | Literal | 6.07 (1.25) | 5.43 (1.46) | 5.00 (1.85) |
| 5: Why get a vaccine if it isn't 100% effective? | *Raincoat* | 5.86 (1.16) | 5.60 (1.41) | 5.24[b] (1.69) |
| | *Seatbelt* | 5.88 (1.23) | 5.38 (1.39) | 4.90 (1.61) |
| | Literal | 5.69 (1.14) | 5.18 (1.49) | 4.35[b] (1.93) |

[aa] indicates this value significantly differs from the other two values within a cell, $p < .05$

[b] indicates these two values within a cell significantly differ, $p < .05$

This analysis suggests that certain metaphors may help or hinder communications about a particular topic. For example, using a *bank* or *castle* metaphor to explain how vaccines work may make a message easier to understand, though it does not appear to impact how informative or persuasive the message seems in relation to a comparable literal message. Using a *war* metaphor involving mobilization by a country under attack to explain why people should take a vaccine even if they are personally at low risk for the illness appears to be particularly ineffective, eliciting lower understandability and informativeness ratings. We offer additional evidence for the efficacy of certain metaphors in the linguistic analysis section below. Overall, however, the explicit ratings data indicate that people's evaluations of the messages were only slightly impacted by the presence of a metaphor. For most of the comparisons, participants tended to perceive the metaphor-enriched messages as similarly understandable, persuasive, and informative as the messages that did not include an extended explanatory metaphor.

## (How) Did overall vaccine attitudes change from pre- to posttest by condition?

We used a repeated-measures ANOVA comparing attitudes towards vaccines before and after participants read and rated the messages (VAM-pretest vs. VAM-posttest). Whether the participant received explanatory metaphor messages or literal messages was included as a between-subject factor (note that this analysis does not distinguish between which of the two metaphorical messages a participant received if they were in the metaphor condition). Previous research—including our own pilot testing—has shown that a variety of individual differences and demographics are associated with attitudes towards vaccines. Therefore, we analyzed age, self-reported science background, generic conspiracist beliefs, trust in institutions, and political ideology as covariates in the model. Two individuals left certain demographic variables blank and were therefore excluded from this analysis.

Overall vaccine attitudes became significantly more favorable from pretest ($M = 4.98$, $SD = 1.22$) to posttest ($M = 5.23$, $SD = 1.24$), $F(1, 292) = 19.27$, $p < 0.001$. However, there was no interaction between vaccine attitudes (pre- vs. posttest) and condition, $F(1, 292) = 0.059$, $p = 0.81$, as illustrated in Fig 1. These findings suggest that simply taking part in a study reading healthcare communications led to more favorable/accurate attitudes towards vaccines, whether or not extended explanatory metaphors were included in the messaging.

Interestingly, a significant within-subject effect was found in the interaction between vaccine attitudes (pre- vs. posttest) and trust in institutions, $F(1, 292) = 7.17$, $p = 0.008$. A similar effect was found for generic conspiracy beliefs, though this effect was marginal, $F(1, 294) = 5.05$, $p = 0.05$. People whose questionnaire responses indicated they had lower trust in institutions or more generic conspiracy beliefs showed a greater increase in vaccine attitude scores from pre to posttest compared to people with higher trust in institutions or fewer conspiracy beliefs. However, this is likely a result of lower pretest baseline scores for these low-trust and high-conspiracy individuals.

Three covariates independently predicted vaccine attitudes collapsing across pre- and posttest: (a) generic conspiracist beliefs ($F(1, 292) = 154.86$, $p < 0.001$), (b) political ideology ($F(1, 292) = 118.96$, $p < 0.001$), and (c) trust in institutions ($F(1, 292) = 12.75$, $p < 0.001$). Endorsing conspiracy beliefs, conservative political ideology, and lower trust in institutions were all associated with less favorable vaccine attitudes (i.e., lower scores on our measure). This is consistent with previous research on vaccine hesitancy [5–9] and provides additional empirical support for those findings while highlighting the validity of our adapted vaccine attitude measure.

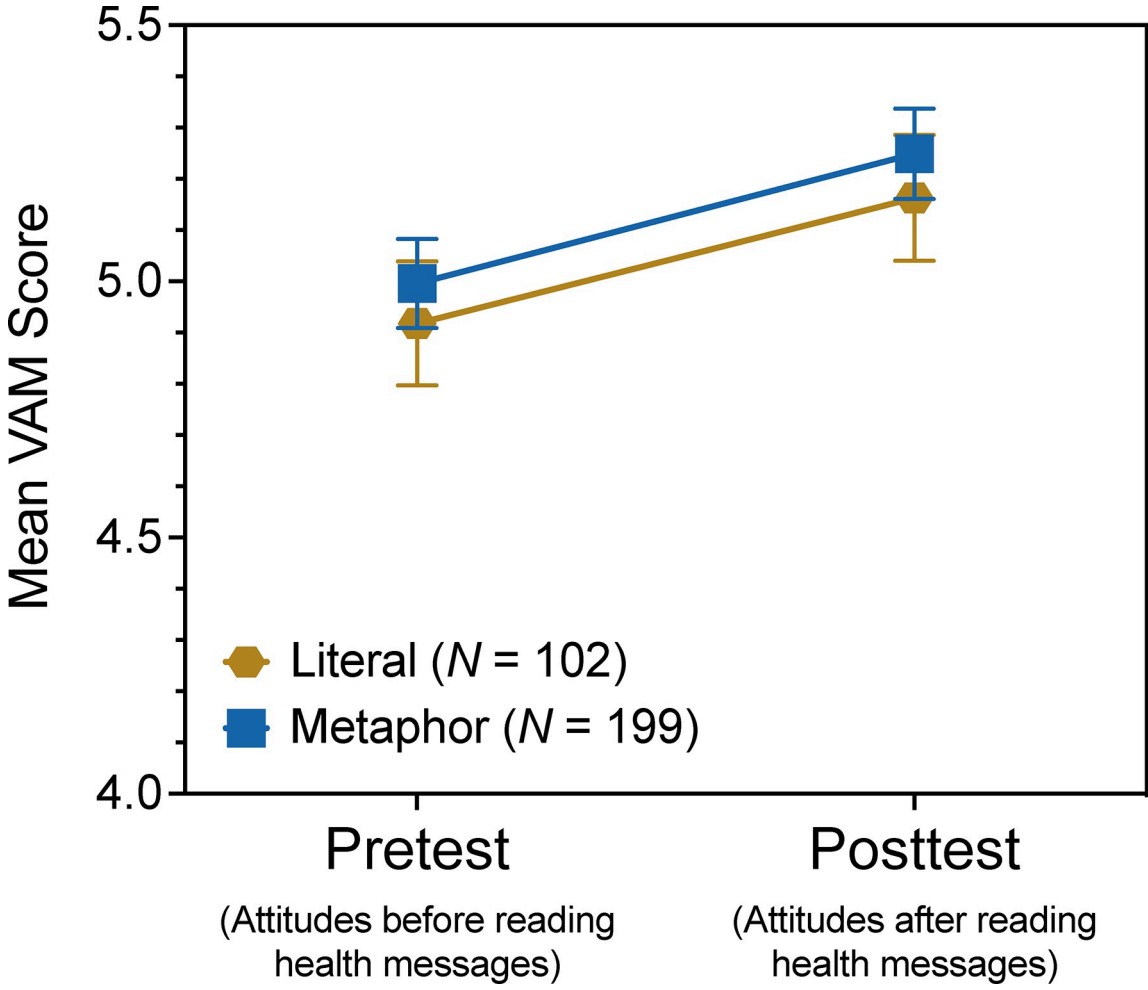

**Fig 1. Mean pretest and posttest Vaccine Attitude Measures by condition.** Error bars represent standard errors of the mean. Data for all 301 participants is shown.

### Exploratory analyses by vaccine attitude sub-category

We conducted an identical repeated-measures analysis for each of the eight VAM sub-category scales, including all of the same covariates as before. This enabled us to explore whether the messages led to differential impacts on participants' attitudes toward certain vaccine attitude items since some of the sub-categories corresponded directly to the content of the messages (e.g., sub-category 2, on "natural immunity," corresponds to vaccine Question 2 "Is 'natural immunity' better than the immunity provided by a vaccine?").

Overall, most of the same general patterns were obtained for the sub-category scales as for the total VAM analysis, though the findings are noisier due to increased variance in the smaller sub-scales. However, these data do suggest that health messages may have been particularly effective at targeting the specific content included in the messages. VAM scores increased significantly from pre- to posttest for sub-category scales 2 ($F(1, 292) = 16.95$, $p < 0.001$), 3 ($F(1, 292) = 22.18$, $p < 0.001$), and 5 ($F(1, 292) = 12.65$, $p < 0.001$). The effect for subscale 4 was similar, though this test did not quite reach significance ($F(1, 292) = 3.88$, $p = 0.05$). Interestingly, VAM scores *decreased* between pre- and posttest for sub-category scale 1, though not significantly ($F(1, 292) = 3.33$, $p = 0.069$), which is a finding we unpack in the General Discussion.

Each of these sub-scales was associated with one of our target questions (see Tables 2 and 3). VAM scores did not significantly differ between pre- and posttest for sub-category scales 6, 7, and 8, though the numeric trends were in line with the increase observed in the other sub-scales. Sub-scales 6–8 tapped beliefs and attitudes that were not explicitly addressed in our health messages (i.e., personal attitudes towards vaccines, attitudes towards childhood vaccinations, and vaccine conspiracy beliefs). Future work is needed to develop messages to target these issues directly.

Whether or not the message included an explanatory metaphor was not a significant factor for any of the subscales, with one notable exception: For sub-category scale 2 ('natural immunity'), scores increased significantly more between pre- and posttest for participants in the metaphor condition than the literal condition, $F(1, 292) = 16.95$, $p = 0.036$. See Fig 2. In exploratory research like the current study, the possibility of false positives must always be considered. However, this result may also indicate that metaphors are especially effective for communicating about this particular issue of natural immunity. Current research underway in our labs is further testing this intriguing possibility.

## Linguistic analyses

As noted above, the current study aimed to advance traditional procedures in metaphor research in that we asked our participants to report how they would respond to a friend asking the target questions ("How would you respond to a friend who asked you. . .?"). Participants filled in a free-text response box, into which they could type without a time or character limit. This elicitation method, while not uncommon in social science and survey-based research (e.g., [54]), has not been used in prior metaphor framing research (although [22] used a related but more abstract 'imagine a situation' measure). This technique was designed to obtain data to triangulate the attitude measures of the efficacy of public health messaging. It also allowed us to gauge the extent to which the different explanatory metaphors participants read may have been memorable or 'sticky' for them (in the sense of engagement or involvement). We were interested to see whether metaphors participants had read were reused by them when they answered the vaccine-related concerns of their purported friend. The "explain to a friend" prompt also allowed us to investigate *how* the different explanatory metaphors were reused. Their free response data allowed us to touch upon social communication in interactions with friends and family in decisions about vaccinations. This dovetails with reports like the UK's SAGE Working Group on Vaccine Hesitancy [9], where the authors include individual and group influences as one of three groups of determinants of vaccine hesitancy. As we took a first pass through the data, a preliminary posthoc analysis revealed that the answers to the "explain to a friend" prompt also provided novel participant-generated metaphors, which were not a focus of the current study. Although space constraints prevent a full discussion in the current paper, assessment of these data may reveal insightful information about the relationship between the original messages participants read and the new metaphors they constructed themselves to explain vaccines to their friends. We look forward to describing these findings in future reports.

**Did the length of free-text responses (in terms of word tokens) differ depending on condition?.** To address this component of participant responses, we calculated the length of free-text responses in terms of the number of words (tokens) used. The results are displayed in Table 5 and Fig 3.

Overall, participants in the metaphor condition ($M = 140.5$, $SD = 73.6$) used significantly more total words on average (~18%) in their "explain to a friend" responses than those in the literal condition ($M = 119.2$, $SD = 62.7$), $t(299) = 2.51$, $p = 0.013$, $d = 0.31$. This effect was

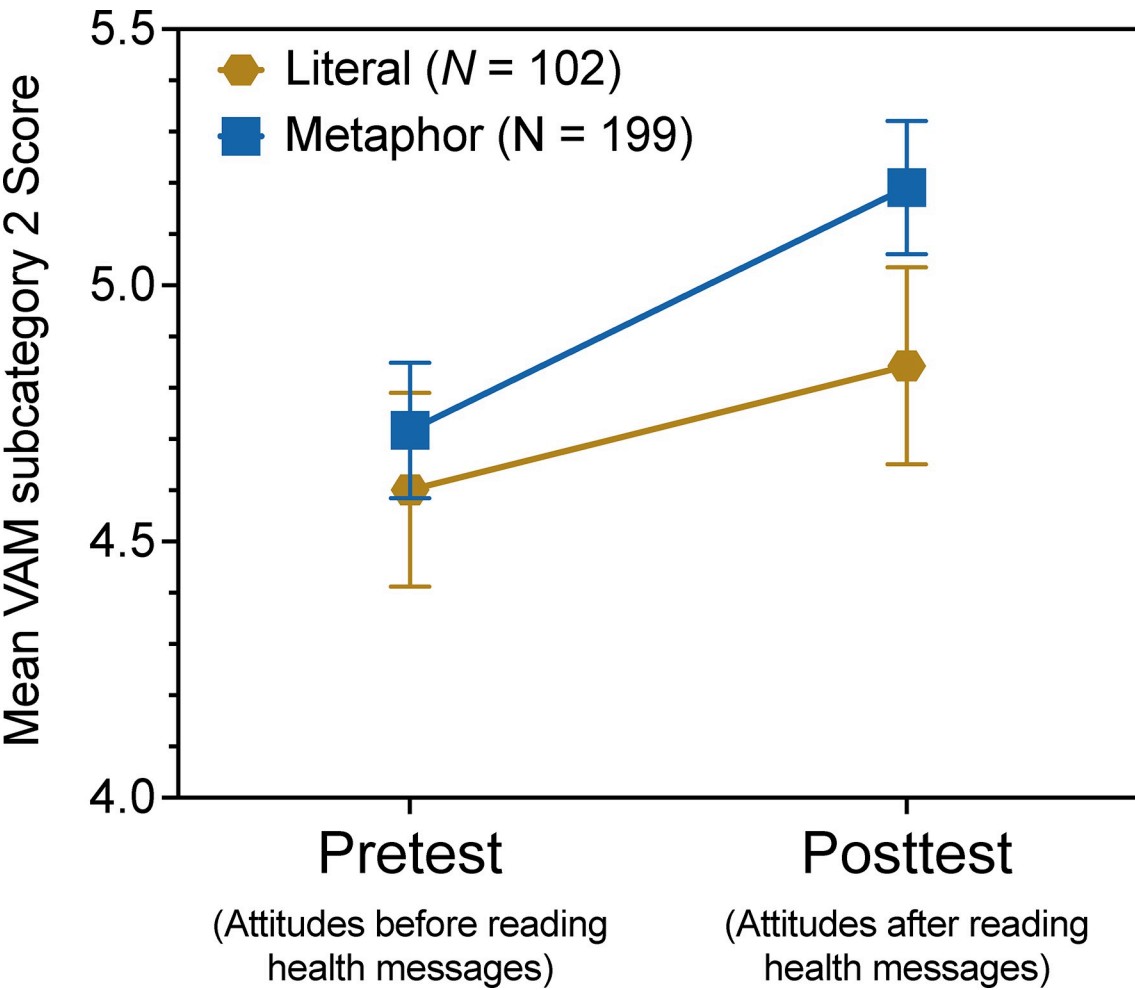

**Fig 2. Mean pretest and posttest VAM subcategory 2 scale scores by condition.** Error bars represent standard errors of the mean. Data for all 301 participants is shown.

**Table 5. Cumulative and mean number of word tokens in free-text responses for each condition and question.**

| Condition | # Tokens | Question 1 | Question 2 | Question 3 | Question 4 | Question 5 | Total |
|---|---|---|---|---|---|---|---|
| Literal (N = 102) | Mean (SD) | 25.3 (16.8) | 22.5 (13.1) | 24.9 (15.2) | 23.6 (15.4) | 22.8 (15.7) | 119.2 (62.7) |
| | Cumulative | 2584 | 2291 | 2541 | 2412 | 2327 | |
| Metaphor (N = 199) | Mean (SD) | 26.8 (16.5) | 28.7 (17.0) | 29.4 (17.7) | 27.6 (17.2) | 27.9 (18.6) | 140.5 (73.6) |
| Metaphor 1 | | Castle (N = 101) | Pilot (N = 101) | Cake (N = 99) | Speed Limit (N = 99) | Raincoat (N = 100) | |
| | Mean (SD) | 28.2 (18.0) | 28.6 (16.3) | 28.6 (17.5) | 25.5 (15.7) | 29.5 (18.4) | |
| | Cumulative | 2848 | 2884 | 2836 | 2521 | 2954 | |
| Metaphor 2 | | Bank (N = 98) | Fire Drill (N = 98) | Video Game (N = 100) | War (N = 100) | Seatbelt (N = 99) | |
| | Mean (SD) | 25.4 (14.8) | 28.9 (17.8) | 30.2 (17.9) | 29.8 (18.4) | 26.3 (18.8) | |
| | Cumulative | 2486 | 2835 | 3020 | 2981 | 2607 | |

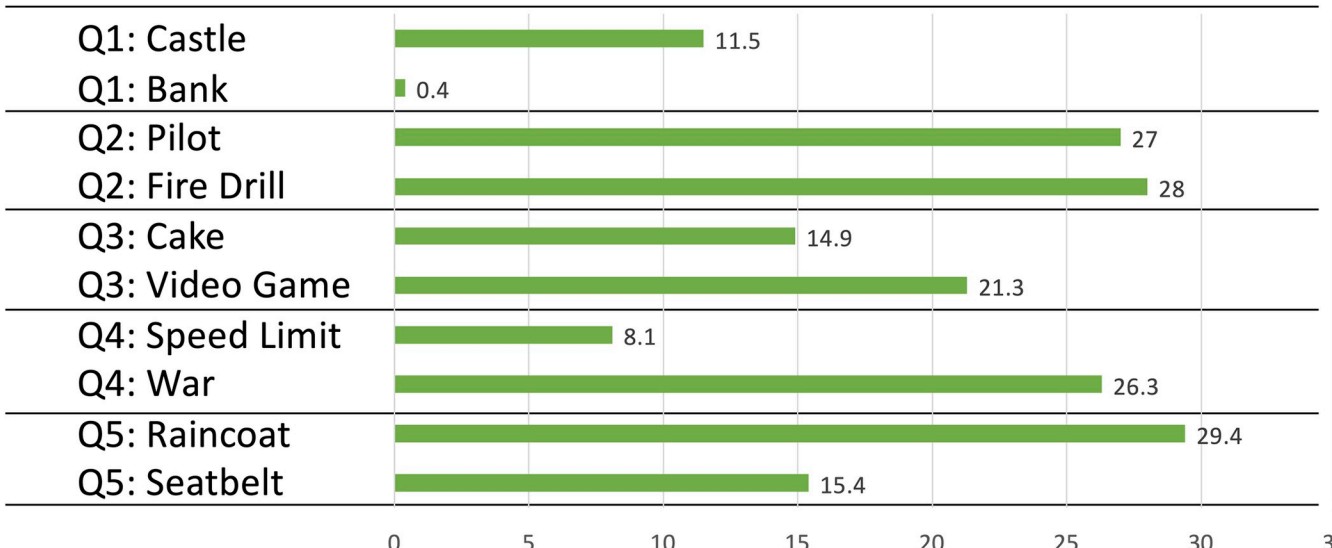

**Fig 3. Percentage difference in mean word production on the "explain to a friend" free response task for each metaphor relative to the comparable literal condition.**

observed fairly consistently across the questions. Participants used significantly more words in the metaphor condition than the literal condition for Question 2 ($t(299) = 3.27$, $p = 0.001$, $d = 0.40$), Question 3 ($t(299) = 2.19$, $p = 0.029$, $d = 0.27$), Question 4 $t(299) = 1.98$, $p = 0.049$, $d = 0.24$) and Question 5 ($t(299) = 2.39$, $p = 0.018$, $d = 0.29$), though not for Question 1 ($t(299) = 0.73$, $p = 0.466$, $d = 0.09$). See Table 5.

Breaking the word counts down by metaphor is illuminating. As illustrated in Table 5 and Fig 3, compared to participants who read literal passages, those who received the metaphorical messages used more words on average for all their explanations to a friend. This difference was negligible (0.4%) when comparing those who received the *bank* metaphor to those who read the literal passage for Question 1. Overall, however, the average word count for "explain to a friend" answers in eight out of the ten metaphor conditions was at least 10% higher than for the corresponding literal conditions. And in four of the metaphor conditions (*pilot*, *fire drill*, *war*, and *raincoat*), participants used more than 25% more words compared to those who received comparable literal messages.

**Table 6. Percentage of reuse of each explanatory metaphor in "explain to a friend" responses.**

| Reuses metaphor | | N reuses | % reuses |
|---|---|---|---|
| Q1 | *Castle* | 22 | 21.78 |
| | *Bank* | 23 | 23.47 |
| Q2 | *Pilot* | 8 | 7.92 |
| | *Fire Drill* | 8 | 8.16 |
| Q3 | *Cake* | 9 | 9.09 |
| | *Video Game* | 16 | 16.00 |
| Q4 | *Speed Limit* | 9 | 9.09 |
| | *War* | 7 | 7.00 |
| Q5 | *Raincoat* | 27 | 27.00 |
| | *Seatbelt* | 18 | 18.18 |

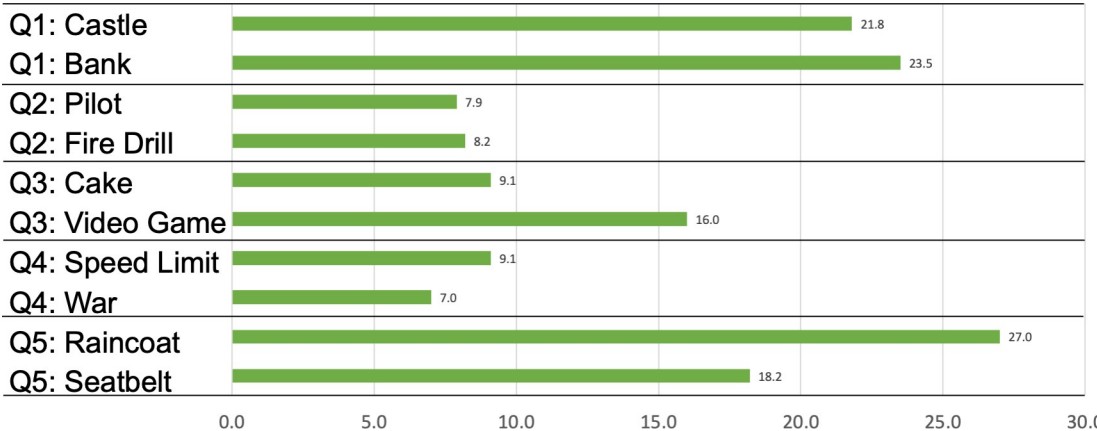

**Fig 4. Percentage of reuse of each explanatory metaphor in "explain to a friend" responses.**

**To what extent were the metaphors participants read then reused in their explanations to a friend's answers?.** We also coded the "explain to a friend" responses in terms of whether or not the explanatory metaphor contained in the passage that each participant read was then reused by the participants in the explanations they said they would give to their friends. All data were double-coded by two research assistants who, after a 45-minute training session involving 5% of the data, proceeded with individual coding. Inter-rater reliability (IRR) was calculated using all data across the five metaphor stimuli using Cohen's Kappa, revealing a high degree of agreement in binary *Yes-No* metaphor reuse ($M$ Cohen's Kappa = 0.94 across the five stimuli). The two coders met to resolve coding disagreements through additional discussion until they achieved complete agreement on all codes ($M$ Cohen's Kappa = 1.00 across the five stimuli). Table 6 and Fig 4 show the outcomes of the coding.

Five of the metaphors were reused by more than 10% of the participants—*raincoat*, *bank*, *castle*, *seatbelt*, and *video game*—indicating that these metaphors were likely particularly salient for participants as they responded to the "explain to a friend" prompt. Three metaphors were reused to an even greater extent—by more than 20% of participants. These especially salient metaphors were *castle* (22%), *bank* (23%), and *raincoat* (27%).

The metaphors used in response to Questions 1 ("How do vaccines work?") and 5 ("Why get a vaccine if it isn't 100% effective?") were reused most often. This might be because of the topic of the questions (e.g., how and why), or because of an inherent attribute of the metaphors (more memorable or"sticky"), or due to the interaction between the two. The two metaphors for Question 1 were reused to a similar extent, whereas the *raincoat* metaphor was reused more than the *seatbelt* metaphor in relation to Question 5. The metaphors for Question 2 were reused less often. When considering which specific metaphors were retained and reused by participants, we also found it interesting to look at *how* they were used.

**How are the metaphors reused in free-text answers?.** The first public health messaging question asked was "How do vaccines work?" The metaphors used in the explanatory passage were those of a *castle* (22 reuses) and a *bank* (23 reuses). For both metaphors, participant reusers can be divided between those who mention two lines of defense or security (13/22 for *castle*, 9/23 for *bank*) and those who only mention one (9/22 for *castle*, 14/23 for *bank*). This may be because the idea that vaccines help fight viruses and defend the body is conventional and popular in public health messaging, but the idea of two unique lines of defense may be less familiar to some people.

Examples of reuse with one line of defense can be seen in the following extracts from the data: "Vaccines work by training the body to recognize foreign invaders that can cause infections. The vaccine helps your body protect against them and fight them" and "Vaccines work like an alarm system to alert your body when a virus is trying to get in and to stop it." Examples of reuse with two lines of defense can be seen in the following two extracts from the data: "Vaccines work like a medieval castle where you have multiple lines of defense including archers to protect the virus from getting in the castle and elite troops that help drive the virus out if it penetrates the castle walls (your body)" and "Getting a vaccine is like getting a comprehensive security system for your body, with two layers of defense. First, it has preventative measures, like security cameras, always on the lookout. But if one of the cameras fails and someone unwanted gets through, a team of trained security guards deals with them. This is how a vaccine might deal with a virus getting through."

Interestingly, a few participants reused the metaphors in the explanatory passage while not getting the details of how vaccines work right, for example: "Your body has cells that fight like warriors. A virus tries to enter your body and if so these warriors (cells) try to fight it off, it if it can't that is when the vaccine will also help as a back up."

None of the *castle* reuses mention boosters, but three *bank* reuses do. We speculate that one possible reason for this may be because the idea of an "update" is more consistent with security systems than weapons: "They work like high tech security cameras that are watching and protecting you from intruders and can catch them before they make a severe impact and the boosters are like an upgrade to new features, or information that can help protect even more from getting in."

In relation to Question 3, the *video game* metaphor was reused 16 times, which included several instances of resistance to the metaphor, for example, using the metaphorical scenario in a way that undermines the role of vaccines, as in the following extract from the data: "Safe compared to what? If you mean are rapidly developed vaccines tested enough to be proven safe then the answer is no. Editing vaccines isn't like adding powers to a video game character. The effects of each new addition have to be thoroughly tested over time. If the disease is so severe that it threatens society, then cutting time to try and save lives may be warranted. Otherwise, vaccines shouldn't be rushed or announced save without proper research to back up those claims" and "The article says yes and it's just like a video game patch. However, we have seen that certain vaccines have worse side effects than others." and finally, "No, they are not, because they have not been adequately tested. Just think of a video game, where the developer rolls out a new feature, sure that it will work as they planned. But, soon enough, the new feature begins acting buggy, and causing the whole game to lag and no longer work as it should. Then, the developer has to panic, and work around the clock to try and find a solution to the problem they inserted, all while trying to save their user base."

Question 5 asks, "Why get a vaccine if it isn't 100% effective?" The *raincoat* metaphor attracted 27 reuses, with participants using different aspects of the scenario, corresponding to different causes of increased risk of infection, for example, deficient raincoats vs. very bad weather. The following extracts from the data illustrate this range of reuses: "It is better to get protected 50% than 0% percent. Just like you would still wear a raincoat if even if [sic] it wasn't the best raincoat ever, you should get a vaccine because it may protect you enough to keep you out of the hospital. Also it will protect others." and, "It provides a high degree of protection against you becoming seriously ill. Would you rather wear a raincoat or walk around unprotected during a thunderstorm?" In the latter example, the use of a rhetorical question and the contrast between wearing a raincoat and being "unprotected during a thunderstorm" arguably have stronger evaluative and emotional associations than the wording used in the original stimulus text.

For some of the metaphors, reuses built upon and went beyond the stimulus texts in terms of the level of detail or main focus in the answers to the friend scenario. These tended to be by high reusers, for example, in response to the *fire drill* metaphor used to illustrate the explanation given to Question 2: "It is not! Think of vaccine immunity like training in a fire drill. You're in a controlled environment where you're not at high risk if something goes wrong. If you were in a real fire situation, just like trying natural immunity, if you panic or do something wrong you could get seriously hurt, or worse. But with the safety of a fire drill where there is no actual fire, you can make mistakes. Similarly, with a vaccine, your body is being trained to deal with infection in a safe, controlled manner." There was a similar pattern in response to the *cake* metaphor used in the explanatory passage for Question 3: "I would respond that I think we have to think about how quickly the virus can mutate and we have to think about how if we don't have some ready now and start getting people vaccinated that it'll mutate worse and worse. We should look at it like a cake, the vaccine is the batter that is already pre-made and you come in to get a personalized message if more and more people came in and the batter wasn't ready, the cake makers would be behind and as new mutations come in, they wouldn't be able to control it and it would cause chaos." Finally, we saw the same pattern of using the metaphor as a jumping-off point and going beyond it when explaining to a friend in participants who read the explanatory passage with the *video game* metaphor given to illustrate the answer to Question 3: "Quickly developed vaccines are perfectly safe. They're not made from scratch. Think of older vaccines as a video game, like Minecraft. The framework is already there, already made. When a new vaccine needs to be made, that framework can be drawn upon. It's like making a mod or downloading the latest patch for Minecraft—they don't have to build Minecraft in a rush every time. They just take what exists and improve upon it."

## General discussion

The COVID-19 pandemic has helped illuminate the risks posed to public health by vaccine hesitancy. The development of effective messaging to improve vaccine attitudes may help improve public health outcomes. Metaphors are a promising tool for communicating about complex issues and have previously been shown to bolster persuasion and explanation [17, 18]. Perhaps not surprisingly, scientists, doctors, and public health officials often use metaphors to explain how vaccines work and to address common questions, concerns, and misconceptions. However, the effectiveness of these metaphors has not been convincingly demonstrated empirically (though some researchers have begun to address this issue, e.g., [37]). In the current study, we investigated the impact of a range of explanatory metaphors on vaccine attitudes. We also included a new methodological tool in metaphor research with the goal of opening a window to view the role of social communication in assessing the impact of metaphors.

For participants in our study, communicating with extended explanatory metaphors did not make a vaccine health message any more or less effective than communicating via a comparable literal message overall. The few exceptions we did find, though, may be useful for researchers and public health communicators who need to make choices about particular metaphors in the future. For example, our quantitative findings suggest that invoking a *war* metaphor involving a national mobilization scenario might be particularly ineffective in explaining why low-risk individuals should be vaccinated. Given that there is a wide range of possible available metaphors, using a different one is a simple choice. In contrast, the *castle* metaphor, which involves a different war-related scenario, was found to be potentially useful in communicating how vaccines work. This highlights the importance of considering specific scenarios and their associated narratives and structural entailments in developing metaphorical frames.

Broader domains like WAR encompass a range of different scenarios that may be more or less effective when used as metaphors to explain particular target domains.

Our exploratory analysis has also revealed that using *pilot training* or *fire drill* metaphors to explain why natural immunity is not inherently superior or different from vaccine-induced immunity may be particularly effective. Future work is needed to replicate these findings, but they speak to the potential promise of certain metaphorical messages over others in addressing common misconceptions about vaccines.

Regardless of whether participants read messages with or without metaphors, there was a small but significant increase in favorable attitudes towards vaccines after reading health care messages. This argues for the effectiveness of communications about vaccines and is a positive finding for healthcare messaging in general.

Exploratory analyses of our VAM subscales provided some evidence that vaccine attitudes were especially likely to improve for issues directly addressed by our health messages and may have been less likely to improve for issues we did not cover directly (e.g., vaccine conspiracies and childhood vaccinations; as illustrated in the data presented in Tables 2 and 3). The only VAM subscale that was directly tied to one of our target questions/messages and that showed a decrease from pre- to posttest concerned how vaccines work (subscale 1). This suggests there may be issues with the specific items in this subscale. It is possible the two reverse-coded items were confusing. Vaccines are not technically "designed to target and neutralize viruses that enter the body," nor do they "contain new antibodies designed to deal with infections." But vaccines do essentially function to help the body create new antibodies to target and neutralize viruses, so participants may have interpreted the statements in that way. In future work, we plan to clarify the language in these statements to avoid confusion.

Our findings also replicate previous research showing that specific individual difference factors can be associated with more negative attitudes toward vaccines. These include political ideology and trust in institutions. Importantly, though, controlling for these factors did not eliminate the impact of the health messages. This demonstrates that brief educational interventions on this subject can be effective, even for individuals who show evidence of vaccine-hesitant beliefs at the start. Again, this is a positive finding for the efficacy of healthcare messaging in general.

One innovative aspect of the present study was the inclusion of free-text responses where participants thought about and reported how they would answer each target question if posed by a friend (cf. [22]). Our linguistic analyses revealed people who read messages containing metaphors produced about 18% more words on average in free-text responses than participants who received the corresponding literal messages. Metaphors, then, seem to provide additional vocabulary and imagery that can be exploited in social relationships—in this case, communicating with a friend. This is particularly relevant given the importance of social networks in decisions about vaccinations [9].

Notably, and somewhat unexpectedly, we found that most participants did *not* reuse the metaphors they had read. The higher word count produced in the metaphor condition suggests that exposure to metaphors results in participants being more productive, or fluent, in producing explanations. Thus, even though providing explanatory metaphors did not directly lead to the reuse of those metaphors, we still saw evidence of a critical metaphor framing effect. On the basis of these data, we posit that exposure to explanatory metaphors may help people organize their understanding of vaccines in a way that facilitates and enriches their subsequent (at least, immediately proximal) communications about the issue. Our study illustrates the potential of this methodology for obtaining insights that go beyond the common quantitively oriented survey techniques that have so far been the primary means of investigating the effects of metaphors on attitudes and decision-making.

Like social communication, individual variation in metaphor use has not received much attention in previous work either. Our linguistic analysis identified a small group who were very competent at reusing the metaphors they read. In addition to this, other participants generated their own explanatory metaphors that were not included in the health messages they received. Our ongoing research is investigating these issues further since we believe they warrant additional attention.

There are, of course, multiple limitations in a study such as ours. Our participants were limited to North American users of Amazon's Mechanical Turk (MTurk). While this platform is widely used in social science research and has proven to be a reliable sampling resource [55, 56], some scholars have recently voiced concerns about the presence of bots and other data quality issues [57]. While we implemented several best practices designed to mitigate these problems (e.g., including an attention check question and using CloudResearch to recruit participants, which pre-screens MTurk users and has been shown to yield higher quality data [40, 41]), future work should aim to replicate these findings with alternative populations. Relatedly, the lack of geographic diversity means our results may be of limited generalizability. As we alluded to above, individual differences may change the way people understand or perceive the metaphors, with downstream consequences for their vaccine attitudes. Although we piloted every aspect of the design, the questions organizing our fictional messaging campaign may not be formulated in a way that resonates with many outside "WEIRD" populations [58]. Future work should extend these methods to other languages and cultures. Additionally, we did not consider the full set of possible metaphors or uses of metaphor in the context of vaccine communication, which limits the generalizability of our findings. Other research has found that using emotionally charged metaphors for a virus (e.g., *beast*, *riot*, *army*)—rather than for vaccines—may increase the willingness to get vaccinated, though more research is needed on this topic [36].

While our adapted Vaccine Attitudes Measure showed evidence of reliability and validity, we have documented possible issues with certain statements that may have confused participants. We are refining the measure for use in future work. Similarly, our linguistic analysis—while having very high inter-rater reliability—comprises a novel use of a borrowed methodology in metaphor research, and its use should, therefore, be viewed as exploratory in nature in this initial study. Additionally, the fact that we did not find clear effects for metaphors over comparable literal passages on vaccine attitudes may not be because there are none but because of the topic of vaccinations itself—which, at the point of data collection, was something many participants would have strong views about due to the timing of this study in the COVID pandemic. Our ongoing work is investigating these research questions with different topics and in both the U.K. and the U.S. contexts. Future work could also usefully consider differences in responses to metaphors depending on whether they emphasize the benefits of vaccination for the individual (e.g., *castle*, *bank*, *raincoat*, and *seatbelt* metaphors) vs. benefit for others (e.g., *speed limits* and *war* metaphors). For example, a randomized control trial on the effects on U.K. adults of different types of vaccine information strategies during the COVID-19 pandemic found that, for strongly vaccine-hesitant participants, information about the personal benefit of vaccination reduces hesitancy to a greater extent than information about collective benefits [59].

Notwithstanding these limitations, we have shown that (1) Brief health messaging passages have the potential to improve attitudes towards vaccines, (2) Explanatory metaphors neither enhance nor reduce this effect relative to comparable literal passages, but (3) Explanatory metaphors may be more helpful than comparable"literal" language in facilitating further social communication about vaccines. Fully addressing vaccine education and hesitancy will take more than a simple 10-minute online intervention. However, we believe our study represents

one cost-effective method for systematically generating and testing messages that healthcare workers and everyday citizens might use in interpersonal, on-the-ground communications. We hope that these findings will both aid and inspire future research on the functions of metaphor in explanation in general and vaccine hesitancy in particular.

## Acknowledgments

We would like to thank K. Cook, E. Fell, and D. Reagan for their assistance with coding the linguistic data and formatting the paper and E. Fell for her invaluable work preparing the revised and final versions.

## Author Contributions

**Conceptualization:** Stephen J. Flusberg, Alison Mackey, Elena Semino.

**Data curation:** Stephen J. Flusberg, Alison Mackey, Elena Semino.

**Formal analysis:** Stephen J. Flusberg, Alison Mackey, Elena Semino.

**Funding acquisition:** Elena Semino.

**Investigation:** Stephen J. Flusberg, Alison Mackey, Elena Semino.

**Methodology:** Stephen J. Flusberg, Alison Mackey, Elena Semino.

**Project administration:** Stephen J. Flusberg.

**Writing – original draft:** Stephen J. Flusberg, Alison Mackey, Elena Semino.

**Writing – review & editing:** Stephen J. Flusberg, Alison Mackey, Elena Semino.

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
