## [Decision Letter · Decision Letter 0]

14 Jun 2023

PONE-D-23-09747Seatbelts and Raincoats, or Banks and Castles: Investigating the Impact of Vaccine MetaphorsPLOS ONE

Dear Dr. Flusberg,

Thank you for submitting your manuscript to PLOS ONE. After careful consideration, we feel that it has merit but does not fully meet PLOS ONE’s publication criteria as it currently stands. Therefore, we invite you to submit a revised version of the manuscript that addresses the points raised during the review process.

Specifically, the Reviewers pointed out sever issues demanding more details in the manuscript (including method description and reporting results). On top of that, please move details related to your study design from section 1.3 ("The current study")  to the section 2 ("Method"). Instead, it would be valuable to elaborate on your research question and the corresponding research gap in section 1.3.

We look forward to receiving your revised manuscript.

Kind regards,

Wojciech Trzebinski, Ph.D.

Academic Editor

PLOS ONE

Journal Requirements:

This work was funded by UK Research and Innovation, grant numbers: ES/R008906/1 and ES/V000926/1 to ES

Reviewers' comments:

Reviewer's Responses to Questions

**Comments to the Author**

1. Is the manuscript technically sound, and do the data support the conclusions?

Reviewer #1: Yes

Reviewer #2: Yes

2. Has the statistical analysis been performed appropriately and rigorously? 

Reviewer #1: Yes

Reviewer #2: Yes

3. Have the authors made all data underlying the findings in their manuscript fully available?

Reviewer #1: Yes

Reviewer #2: Yes

4. Is the manuscript presented in an intelligible fashion and written in standard English?

Reviewer #1: Yes

Reviewer #2: Yes

5. Review Comments to the Author

Reviewer #1: This manuscript is a well-written research study assessing vaccine attitudes of study participants following review and rating of metaphorical and literal responses to 5 common vaccine questions. There are some concerns regarding interpretation of the study findings that should be addressed prior to publication of this manuscript.

Major concerns:

1) Regarding section ‘3.2. Explicit ratings of the health messages (page 19 starting at line 333)’

a. Only significant differences are reported of participant ratings of how understandable, informative, or persuasive the metaphor or literal response is. A table summarizing all ratings data (understandable, informative, persuasive) and p-values for each of the 5 questions is much needed in order to be able to interpret the study findings.

b. Within this section, a broad conclusion is made that metaphorical messages are similar to literal messages, even though there is data indicating significant differences between metaphor and literal ratings. Page 19, line 339-343: The authors state, “These results indicate that people’s explicit judgments about the efficacy of the messages were roughly equivalent for all the messages targeting a given question, with a few minor exceptions. In other words, participants tended to perceive metaphor-framed messages as similar in terms of effectiveness when compared to comparable literal descriptions that do not include an extended explanatory metaphor.” It is not appropriate to refer to the significant differences found for question 1 and 5 as “similar” with “a few minor exceptions.” Given that the statistical differences are unique to each of the 5 questions that address different aspects of vaccine hesitancy, they should be interpreted separately, and these sentences should be revised.

c. The interpretation of the ratings of the 5 questions is presented as a conclusion in the abstract. Last sentence of abstract and page 35, line 677: “(2) Metaphors neither enhance nor reduce this effect relative to comparable literal passages.” Given the significant differences in ratings between the metaphor and literal messages, this conclusion is incorrect and warrants revision in line with the suggested revisions to section 3.2.

2) In this study, the authors investigated the impact of explanatory metaphors on vaccine attitudes assessed by a Vaccine Attitude Measures tool. A general conclusion was made that “metaphors are comparable to literal passages.” Their conclusion is contradictory to other studies that have shown positive correlations between patient receipt of a vaccine following a conversation with a physician who used a metaphor. This difference in study outcomes was never addressed in the publication and warrants a brief discussion.

Reviewer #2: I enjoyed this article and learned a lot from the authors’ careful study design and relevant findings. I think that this study would be of great interest to PLOS ONE readers, and I hope to see it published. The study was well designed, and I was especially interested in the original part of the study design that asked participants to “explain it to a friend” and answer vaccine-related questions in their own words. The findings are useful in contributing to a better understanding of how audiences and readers engage different types of vaccine-related information and how information and beliefs and decisions influence one another or not.

As I wrote above, I was most interested in the findings related to the authors’ method of asking participants to then answer a similar question as if it were posed by a friend. This participatory design -esque angle to vaccine and other forms of public health communication is original and exciting. I was very interested to learn about if and how participants used the metaphors that had just been modeled for them, and I wanted to read more about the ways that participants, when asked to take the reigns and answer this question for a hypothetical friend, improvised their own forms of vaccine communication. Thinking about Covid vaccination learnings, we learned that one of the biggest indicators that someone would be vaccinated against Covid was how many people they knew who were vaccinated. This correlation suggests that people are influenced a lot by those around them and suggests that lay vaccine communicators might have a lot to teach us about vaccination communication! I would look forward to reading an article about these findings and their implications.

I did have a few smaller suggestions that I think would help the article (1) better engage with vaccine-related scholarship and (2) clear up a few confusions I had about the study’s methods.

LITERATURE REVIEW

This comment isn’t one that I think absolutely needs to be addressed, but I did want to note that I wished the authors had addressed the growing body of work that investigates why people believe the things they believe about vaccines - i.e., the people the research team is trying and persuade with these metaphors.

I was wary that the authors avoided any reference to scholarship in the social sciences and humanities that has examined how people come to their vaccine beliefs and has largely concluded that deficit-based messaging approaches to vaccine communication and persuasion don’t work very well. Research shows us that, in short, the assumption that people are vaccine hesitant because of ignorance or misinformation and that their beliefs can be “corrected” by delivering persuasive and accurate information is limited. Instead, vaccine beliefs are complicated, fluid, and inextricable from their specific, social situations and histories. Changing vaccine hesitancy involves building trusted, long-term relationships with individuals and communities, not necessarily trying to package the right message and find the right messenger. Here I’m thinking of work by Bernice Hausman, Elena Conis, Nicole Charles, JEnnifer Reich, Maya Goldenberg, Heidi Lawrence, Melissa Leach & James Fairhead, Clare Decoteau, and Andrea Kitta.

I’ll add that this broader more contextualized approach to vaccine hesitancy as a complex social problem rather than a problem of individuals who believe the wrong things came, briefly, into mainstream vaccine discourse during the phased distribution of Covid vaccines, a time when there was a lot of talk about how and why, for example, people without health insurance might be vaccine hesitant because they don’t trust the systems that produce vaccines to look out for them (and not because they don’t understand or trust the vaccine, specifically), or why Americans of color would be skeptical of the US healthcare system and hold off on signing up for a new vaccine.

This isn’t to say that studies focused on questions about effective messaging don’t have a place – as I’ve written in the rest of the review, I think there’s so much of value in this study – but especially in a study that is attuned to audience and learning how targeted audience members make sense of and are or are not changed by a message, I would love to see, in the literature review sections, acknowledgement of the humanistic and social science work out there that is investigating and producing really important findings about how people come to their health beliefs and decisions.

METHODS

Recruitment

I wanted to know more about the limitations and reasons for using Amazon’s mTurk as a recruitment tool. Obviously it’s easy to recruit participants through this tool, but are there other, more research-driven reasons the authors chose to use mTurk?

In particular, I think the authors should discuss to what extent mTurk participants are a representative group to draw conclusions about general vaccination feelings, beliefs, and persuade-ability. I’m thinking again of all of the great social sciences and humanistic work that argues that real and effective vaccine communication must be built in relationships over time. That is, communicating scientific facts to people–even in culturally specific and artful ways–doesn’t work because this kind of generalized, mass communication rarely addresses people’s actual concerns, which are grounded in their lives, experiences, personal and collective histories, and social networks. So vaccine communication needs to be localized, iterative, and built into broader networks of relationships and trust. By asking mTurk anonymous participants who are likely churning through digital tasks one after another to share a bit about their responses to vaccine messages, the authors are not aligning themselves with this body of research. So I wondered what scholarship the authors are drawing on to endorse learning about vaccine beliefs and changes in belief in this way, that is, in a way that asks people to type a bit about their vaccine decisions made hypothetically, abstracted from all context, and only in hypothetical terms?

The research design – asking people to take assessments about conspiracy beliefs, vaccine beliefs, etc., and using an attention check to select participants more likely to actually read the prompts – all looked great. But I wanted to better understand how, in light of so much research pointing to vaccine decision-making as complex, situational, and tied to specific, local contexts, this method is useful to generate knowledge that *actually* captures how people are making vaccine decisions.

I also think the Methods section should address how participants were compensated.

Study design

I wondered about the authors’ choice to use similar metaphors to answer the question “How do vaccines work?” Based on existing scholarship on vaccines and metaphors, both of these metaphors would fall into a category of older commonplace metaphors about bodies, vaccines, and disease–metaphors that see each body as a sealed-off, fortress-like entity that defends itself against outside germs. By contrast, a more contemporary metaphor by which many people understand the body and immunity is that of a complex system (flexible immunity) – the body constantly adapts to its environment and grows stronger through encounters with pathogens and other things, like dirt. (Emily Martin’s work on immunity and immune systems metaphors is foundational here, but other scholarship on vaccination and metaphors has taken up and extended this argument. So, it seems like both of the metaphors here rely on the older model of the body - a discrete self that must be defended - rather than this flexible immunity model. I wonder if the authors can speak to this decision?

I wondered why the authors didn’t consider race as a demographic feature that influences vaccine decision making and trust in institutions (p. 20, lines 356 - 359)?

On p 15 lines 247-8, the authors write “After reading one of the three explanations associated with a particular question,participants were asked to consider how the general public would react to the message and to keep this in mind while responding to four questions.” I didn’t understand why the authors asked the participants to imagine how a general public audience would respond to the persuasive messaging. Isn’t the goal of the study to track how individual participants respond to the messaging? And to see how individuals respond to and are affected by the messages? It seems the participants should be reporting on their own impressions–how easy each participant found a passage to understand, how informative each participant found a passage–and not trying to speak for an imagined, general public audience.

Not a point that needs to be addressed, but a possibly generative question I had while reading:

I was interested in how the results about which metaphors were effective and ineffective might reveal more about public understandings of public health more so than the effectiveness of individual metaphors. For example, the metaphors that rely on individualist ideas of health (i.e., do things that maximize your own personal health and minimize risks to your own persona health) were the more clear, salient, and effective metaphors (e.g., Bank metaphor, Castle metaphor, Raincoat metaphor) And the metaphors that asked audiences to think about health as an interdependent, community state, one in which we should make decisions toward a greater social good fell flat (the war metaphor). Do these findings suggest that specific metaphors themselves were more or less clear or do they suggest that metaphors, arguments, policies, etc that align with individualist notions of health make more sense than metaphors, arguments, polities, etc that align with collective approaches to health?

Sentence-level points

Is there a reason to use the word “Interestingly” on p 3 line 53? And “presciently”? (Global vaccine hesitancy was already a big problem in 2019, so more accurate than prescient, I would say).

Typo on p 4 line 87

I had trouble following the last 2 analytical sentences of the section “1.2. Explanatory metaphors in vaccine discourse” (on p. 8, lines 150 -6)

6. PLOS authors have the option to publish the peer review history of their article (what does this mean?). If published, this will include your full peer review and any attached files.

Reviewer #1: No

Reviewer #2: **Yes: **Kari Campeau

---

## [Author Response · Author response to Decision Letter 0]

24 Jul 2023

PONE-D-23-09747

Seatbelts and Raincoats, or Banks and Castles: Investigating the Impact of Vaccine Metaphors

PLOS ONE

Dear Dr. Flusberg,

Thank you for submitting your manuscript to PLOS ONE. After careful consideration, we feel that it has merit but does not fully meet PLOS ONE’s publication criteria as it currently stands. Therefore, we invite you to submit a revised version of the manuscript that addresses the points raised during the review process.

Specifically, the Reviewers pointed out several issues demanding more details in the manuscript (including method description and reporting results). On top of that, please move details related to your study design from section 1.3 ("The current study") to the section 2 ("Method"). Instead, it would be valuable to elaborate on your research question and the corresponding research gap in section 1.3.

****We have now moved this information as requested. 

● A rebuttal letter that responds to each point raised by the academic editor and reviewer(s). You should upload this letter as a separate file labeled 'Response to Reviewers'.

● A marked-up copy of your manuscript that highlights changes made to the original version. You should upload this as a separate file labeled 'Revised Manuscript with Track Changes'.

● An unmarked version of your revised paper without tracked changes. You should upload this as a separate file labeled 'Manuscript'.

We look forward to receiving your revised manuscript.

Kind regards,

Wojciech Trzebinski, Ph.D.

Academic Editor

PLOS ONE

Journal Requirements:

and 

The title page has been corrected.

This work was funded by UK Research and Innovation, grant numbers: ES/R008906/1 and ES/V000926/1 to ES

Please state what role the funders took in the study. If the funders had no role, please state: 

***We have added this information to our cover letter.

No changes necessary, we have shared all of our data to the Open Science Framework repository listed in the paper. If the paper is accepted for publication, we will de-anonymize the link in the final version. 

*****We have added the following statement to the Participants section:

“All participants gave informed consent prior to beginning the study. This study was reviewed and approved under reference FASSLUMS-2021-0576-RECR-2 by the Faculty of Arts and Social Sciences - Lancaster University Management School Research Ethics Committee at Lancaster University, U.K. (FASS-LUMS).”

Reviewers' comments:

Reviewer's Responses to Questions

Comments to the Author

1. Is the manuscript technically sound, and do the data support the conclusions?

Reviewer #1: Yes

Reviewer #2: Yes

2. Has the statistical analysis been performed appropriately and rigorously? 

Reviewer #1: Yes

Reviewer #2: Yes

3. Have the authors made all data underlying the findings in their manuscript fully available?

Reviewer #1: Yes

Reviewer #2: Yes

4. Is the manuscript presented in an intelligible fashion and written in standard English?

Reviewer #1: Yes

Reviewer #2: Yes

5. Review Comments to the Author

Reviewer #1: This manuscript is a well-written research study assessing vaccine attitudes of study participants following review and rating of metaphorical and literal responses to 5 common vaccine questions. There are some concerns regarding interpretation of the study findings that should be addressed prior to publication of this manuscript.

Major concerns:

1) Regarding section ‘3.2. Explicit ratings of the health messages (page 19 starting at line 333)’

a. Only significant differences are reported of participant ratings of how understandable, informative, or persuasive the metaphor or literal response is. A table summarizing all ratings data (understandable, informative, persuasive) and p-values for each of the 5 questions is much needed in order to be able to interpret the study findings.

***We appreciate this helpful suggestion and agree that a table summarizing the ratings data would be useful. We have now added the table (Table 4, pages 20-21). 

b. Within this section, a broad conclusion is made that metaphorical messages are similar to literal messages, even though there is data indicating significant differences between metaphor and literal ratings. Page 19, line 339-343: The authors state, “These results indicate that people’s explicit judgments about the efficacy of the messages were roughly equivalent for all the messages targeting a given question, with a few minor exceptions. In other words, participants tended to perceive metaphor-framed messages as similar in terms of effectiveness when compared to comparable literal descriptions that do not include an extended explanatory metaphor.” It is not appropriate to refer to the significant differences found for question 1 and 5 as “similar” with “a few minor exceptions.” Given that the statistical differences are unique to each of the 5 questions that address different aspects of vaccine hesitancy, they should be interpreted separately, and these sentences should be revised.

***We also appreciate this point and apologize for the confusion. The reason we included the statement that the ratings across stimuli were “roughly” equivalent was because there were only six statistically significant differences out of 45 total post-hoc t-test comparisons (5 questions X 3 stimuli X 3 rating categories). However, following the reviewer’s comment, it is clear to us that this statement obscures over meaningful differences in the data. We have now edited this section of the manuscript for additional clarity, nuance, and accuracy (p. 21) to address the concern, as follows:

“This analysis suggests that certain metaphors may help or hinder communications about a particular topic. For example, using a Bank or Castle metaphor to explain how vaccines work may make a message easier to understand, though it does not appear to impact how informative or persuasive the message seems in relation to a comparable literal message. Using a War metaphor to explain why people should take a vaccine if they are personally at low risk for the illness appears to be particularly ineffective, eliciting lower understandability and informativeness ratings. We provide additional evidence in terms of the efficacy of certain metaphors in the linguistic analysis section below. Overall, however, the explicit ratings data indicate that people’s evaluations of the messages were only slightly impacted by the presence of a metaphor. For most of the comparisons, participants tended to perceive the metaphor-enriched messages as similarly understandable, persuasive, and informative as the messages that did not include an extended explanatory metaphor.

c. The interpretation of the ratings of the 5 questions is presented as a conclusion in the abstract. Last sentence of abstract and page 35, line 677: “(2) Metaphors neither enhance nor reduce this effect relative to comparable literal passages.” Given the significant differences in ratings between the metaphor and literal messages, this conclusion is incorrect and warrants revision in line with the suggested revisions to section 3.2.

***Thank you for pointing this out, and again, we apologize for the confusion. The finding referred to in the abstract was not intended to refer to people’s explicit ratings of the stimuli (as described in Section 3.2). Rather, it was intended to refer to the improvement in overall vaccine attitudes, which was similar for those in the literal and metaphor conditions. The abstract included a point about the ratings, however, and we have we now updated it for accuracy and clarity to address the reviewer’s helpful comment:

“Results showed participants in both conditions rated most messages as being similarly understandable, informative, and persuasive, with a few notable exceptions.”

2) In this study, the authors investigated the impact of explanatory metaphors on vaccine attitudes assessed by a Vaccine Attitude Measures tool. A general conclusion was made that “metaphors are comparable to literal passages.” Their conclusion is contradictory to other studies that have shown positive correlations between patient receipt of a vaccine following a conversation with a physician who used a metaphor. This difference in study outcomes was never addressed in the publication and warrants a brief discussion.

****We appreciate this point very much. We don’t believe we can generalize from our study to all other vaccine communication contexts. As we noted in the introduction, there is limited experimental research assessing the efficacy of metaphors on vaccine uptake (though there has been plenty of speculation). The one exception that we are aware of is the set of studies conducted by Scherer and colleagues (2015), which we referenced in our manuscript. Their study was different to ours in a number of key aspects. For example, they used metaphors to describe a specific virus (the flu) rather than to explain how vaccines work. In their first experiment, they found that describing the flu metaphorically as a beast, riot, army, or weed led to increased expressed willingness to be vaccinated compared to describing the flu literally as a virus. Follow-up experiments found mixed evidence that metaphors for the flu impacted requests to receive an e-mail reminder to get vaccinated. 

To address the reviewer’s helpful comment, in our updated manuscript, we have elaborated on our discussion of this study in the introduction to better highlight the differences with our research (pages 8-9):

“Describing the influenza virus as a “beast” is quite different from the elaborate explanatory metaphors used in COVID-19 discourse, however. For one thing, “beast” is a metaphor for a virus, rather than a metaphor for some aspect of vaccination. For another, “beast” is a subtle, one-off metaphor rather than an extended and elaborated explanatory metaphor. In contrast, the “Cake” metaphor described above is explicitly presented as an explanation for how researchers could develop the COVID-19 vaccines so quickly, and it was extended and developed throughout the text. It is this latter type of explanatory metaphor that we aimed to evaluate in the present study.” 

We have also added to our points on limitations in terms of generalizability in the General Discussion (p. 36):

“Additionally, we did not consider the full set of possible metaphors or uses of metaphor in the context of vaccine communication, which limits the generalizability of our findings. Other research has found that using emotionally-charged metaphors for a virus (e.g., beast, riot, army)—rather than for vaccines—may increase a willingness to get vaccinated, though more research is needed on this topic [33]."

Reviewer #2: I enjoyed this article and learned a lot from the authors’ careful study design and relevant findings. I think that this study would be of great interest to PLOS ONE readers, and I hope to see it published. The study was well designed, and I was especially interested in the original part of the study design that asked participants to “explain it to a friend” and answer vaccine-related questions in their own words. The findings are useful in contributing to a better understanding of how audiences and readers engage different types of vaccine-related information and how information and beliefs and decisions influence one another or not.

***We thank you for the kind words and the deep level of engagement with our paper!

As I wrote above, I was most interested in the findings related to the authors’ method of asking participants to then answer a similar question as if it were posed by a friend. This participatory design -esque angle to vaccine and other forms of public health communication is original and exciting. I was very interested to learn about if and how participants used the metaphors that had just been modeled for them, and I wanted to read more about the ways that participants, when asked to take the reigns and answer this question for a hypothetical friend, improvised their own forms of vaccine communication. Thinking about Covid vaccination learnings, we learned that one of the biggest indicators that someone would be vaccinated against Covid was how many people they knew who were vaccinated. This correlation suggests that people are influenced a lot by those around them and suggests that lay vaccine communicators might have a lot to teach us about vaccination communication! I would look forward to reading an article about these findings and their implications.

***We are most gratified that you appreciated this unique aspect of our methods, and we agree this is a critical and fascinating topic to investigate further. A full treatment along these lines was sadly outside the word limit and scope of the current manuscript, as we wanted to focus mainly on developing and assessing the impact of various explanatory metaphors. As we discuss more below, we view the current study as a first step in a much broader and more expansive project aimed at improving communications surrounding vaccines and vaccine uptake, and the reviewer’s comments have been extremely helpful to us in this respect, so again, thank you.

I did have a few smaller suggestions that I think would help the article (1) better engage with vaccine-related scholarship and (2) clear up a few confusions I had about the study’s methods.

LITERATURE REVIEW

This comment isn’t one that I think absolutely needs to be addressed, but I did want to note that I wished the authors had addressed the growing body of work that investigates why people believe the things they believe about vaccines - i.e., the people the research team is trying and persuade with these metaphors.

I was wary that the authors avoided any reference to scholarship in the social sciences and humanities that has examined how people come to their vaccine beliefs and has largely concluded that deficit-based messaging approaches to vaccine communication and persuasion don’t work very well. Research shows us that, in short, the assumption that people are vaccine hesitant because of ignorance or misinformation and that their beliefs can be “corrected” by delivering persuasive and accurate information is limited. Instead, vaccine beliefs are complicated, fluid, and inextricable from their specific, social situations and histories. Changing vaccine hesitancy involves building trusted, long-term relationships with individuals and communities, not necessarily trying to package the right message and find the right messenger. Here I’m thinking of work by Bernice Hausman, Elena Conis, Nicole Charles, JEnnifer Reich, Maya Goldenberg, Heidi Lawrence, Melissa Leach & James Fairhead, Clare Decoteau, and Andrea Kitta.

I’ll add that this broader more contextualized approach to vaccine hesitancy as a complex social problem rather than a problem of individuals who believe the wrong things came, briefly, into mainstream vaccine discourse during the phased distribution of Covid vaccines, a time when there was a lot of talk about how and why, for example, people without health insurance might be vaccine hesitant because they don’t trust the systems that produce vaccines to look out for them (and not because they don’t understand or trust the vaccine, specifically), or why Americans of color would be skeptical of the US healthcare system and hold off on signing up for a new vaccine.

This isn’t to say that studies focused on questions about effective messaging don’t have a place – as I’ve written in the rest of the review, I think there’s so much of value in this study – but especially in a study that is attuned to audience and learning how targeted audience members make sense of and are or are not changed by a message, I would love to see, in the literature review sections, acknowledgement of the humanistic and social science work out there that is investigating and producing really important findings about how people come to their health beliefs and decisions.

***We agree that the insights provided by this body of work are critical. Again, while a full treatment of this literature is beyond the scope of our article, having thought about the reviewer’s perspective, we also agree it is necessary to discuss the broad, complex contours of these issues. In the updated manuscript, we now reference and discuss some of this work in the beginning of the introduction to better situate the contribution and context of our research (p. 3-4):

“Vaccine hesitancy is a complex phenomenon [2]. It has been associated with many factors, including age, education, mistrust in institutions, engaging with misleading sources online, local and sometimes vaccine-specific personal/family histories [3], and ‘folkloric narratives’ [4–8]. A 2014 WHO report from the Strategic Advisory Group of Experts on Immunization (SAGE) includes three categories of determinants of vaccine hesitancy [9]: (a) ‘contextual influences’ (e.g., religion, culture, politics, media environment); (b) ‘individual and group influences’ (e.g., previous experiences with vaccinations by the individual and their kinship and social groups, immunization as a social norm or as not needed or harmful); and (c) ‘vaccine/vaccination-specific issues’ (e.g., new vaccine, mode of administration, cost, risks vs. benefits). Responding to vaccine hesitancy is therefore also a complex enterprise that goes beyond the provision of ‘accurate’ information [10], whether in public health campaigns or in interactions between healthcare providers and individuals. In this context, then, it is clearly important to investigate the utility of different approaches to vaccine-related communications. The current study investigates the effectiveness of explanatory metaphors in public health messaging about vaccination, and their influence on how individuals perceive communicating about different aspects of vaccinations with members of their own social groups.”

As we discuss further below, we have also expanded our General Discussion in a number of places to highlight additional limitations and opportunities for future research that intersect with more interpersonal approaches to vaccine communications. 

METHODS

Recruitment

I wanted to know more about the limitations and reasons for using Amazon’s mTurk as a recruitment tool. Obviously it’s easy to recruit participants through this tool, but are there other, more research-driven reasons the authors chose to use mTurk?

In particular, I think the authors should discuss to what extent mTurk participants are a representative group to draw conclusions about general vaccination feelings, beliefs, and persuade-ability. I’m thinking again of all of the great social sciences and humanistic work that argues that real and effective vaccine communication must be built in relationships over time. That is, communicating scientific facts to people–even in culturally specific and artful ways–doesn’t work because this kind of generalized, mass communication rarely addresses people’s actual concerns, which are grounded in their lives, experiences, personal and collective histories, and social networks. So vaccine communication needs to be localized, iterative, and built into broader networks of relationships and trust. By asking mTurk anonymous participants who are likely churning through digital tasks one after another to share a bit about their responses to vaccine messages, the authors are not aligning themselves with this body of research. So I wondered what scholarship the authors are drawing on to endorse learning about vaccine beliefs and changes in belief in this way, that is, in a way that asks people to type a bit about their vaccine decisions made hypothetically, abstracted from all context, and only in hypothetical terms?

The research design – asking people to take assessments about conspiracy beliefs, vaccine beliefs, etc., and using an attention check to select participants more likely to actually read the prompts – all looked great. But I wanted to better understand how, in light of so much research pointing to vaccine decision-making as complex, situational, and tied to specific, local contexts, this method is useful to generate knowledge that *actually* captures how people are making vaccine decisions.

***We appreciate these important and thoughtful points. We used mTurk for several reasons. As you indicated, it makes recruiting a large sample of research participants fast and easy, as well as cost effective. It has also been shown to be a reliable platform for recruiting research subjects who are more diverse than most convenience samples that were available to us (e.g., undergraduate students taking psychology and linguistics courses). Also persuasive in our decision to use the platform is the fact that a number of key findings in the social and behavioral sciences have been successfully replicated there. 

While some scholars have highlighted issues with data quality on mTurk, including both inattentive subjects and the presence of bots (i.e., fake participants), we took several steps to ensure our data would be reliable. This includes, as the reviewer mentions, using an attention check question at the start of the study as well as our use of CloudResearch to recruit participants. CloudResearch is an online platform that interfaces with mTurk and uses a variety of validated methods to increase the quality of participants who are recruited to complete each study. Several recent studies have shown that using CloudResearch leads to significantly higher quality data. To make sure this is clearer in the paper, we have added several references and included an expanded discussion of this issue in the limitations section of the General Discussion in the updated manuscript (pages 35-36):

“While this platform is widely used in social science research and has proven to be a reliable sampling resource [52,53], some scholars have recently voiced concerns about the presence of bots and other data quality issues [54]. While we implemented several best practices designed to mitigate these problems (e.g., including an attention check question and using CloudResearch to recruit participants, which pre-screens MTurk users and has been shown to yield higher quality data [36,37]), future work should aim to replicate these findings with alternative populations.” 

We also agree with the broader point made – that fully addressing vaccine education and hesitancy will take more than a simple 10-minute online intervention. However, as we noted earlier, we view our current study and manuscript as a first step towards the development of more effective communication tools that could become part of the “localized, iterative” conversations you describe that are “built into broader networks of relationships and trust.” We believe our study represents one cost-effective way to systematically generate and test messages that healthcare workers and everyday citizens might use in these more interpersonal, on the ground, communicative contexts. We have added a brief point to this effect in the General Discussion section in the updated manuscript (p. 37).

I also think the Methods section should address how participants were compensated.

***We have now added compensation information to the Participants section in the updated manuscript (participants were paid $3). 

Study design

I wondered about the authors’ choice to use similar metaphors to answer the question “How do vaccines work?” Based on existing scholarship on vaccines and metaphors, both of these metaphors would fall into a category of older commonplace metaphors about bodies, vaccines, and disease–metaphors that see each body as a sealed-off, fortress-like entity that defends itself against outside germs. By contrast, a more contemporary metaphor by which many people understand the body and immunity is that of a complex system (flexible immunity) – the body constantly adapts to its environment and grows stronger through encounters with pathogens and other things, like dirt. (Emily Martin’s work on immunity and immune systems metaphors is foundational here, but other scholarship on vaccination and metaphors has taken up and extended this argument. So, it seems like both of the metaphors here rely on the older model of the body - a discrete self that must be defended - rather than this flexible immunity model. I wonder if the authors can speak to this decision?

***These are excellent points. Our selection of metaphors for all of the questions was guided by the authentic, real-world metaphors we have observed in COVID-19 vaccine discourse over the past few years. For the question of “How do vaccines work?”, we generally observed these sealed-off, fortress-like metaphors. While the “Castle” and “Bank” metaphors have similar entailments because of this shared domain structure, they do differ in other respects that we felt had the potential to impact our results. For example, for fans of fantasy stories like Lord of Rings and Game of Thrones, these metaphors might resonate more than for others. Research suggests that one’s affinity and interest in a metaphorical source domain might affect how deeply one processes a metaphor. Having said this, we fully agree that using an alternative method to generate maximally distinct metaphors is also warranted, and that is precisely what we intend to do in future research. We appreciate the reference to Emily Martin’s work, which looks very promising in this regard. We now include an additional sentence about this in our General Discussion, speaking to the limitations of our metaphor stimuli (p. 36): 

“Additionally, we did not consider the full set of possible metaphors or uses of metaphor in the context of vaccine communication, which limits the generalizability of our findings.” 

I wondered why the authors didn’t consider race as a demographic feature that influences vaccine decision making and trust in institutions (p. 20, lines 356 - 359)?

***There are a number of reasons for why we did not consider race in our analyses in the current paper. The first is the complexity of the construct, particularly in light of indications that race may be related to vaccine hesitancy for a wide variety of reasons. Participants self-reported their racial identity, and this data turned out to be much more complex than a multiple-choice answer (as you can see if you examine our data file on the Open Science Framework). Since we did not have a priori predictions about how race might interact with our findings, we did not have any clean way of coding this demographic data reliably. Another complication is that race is often confounded with other factors that affect vaccine hesitancy and trust in institutions that we did look at, such as political ideology. In sum, we view race as another important topic for future research that is outside the scope of the current manuscript. We are committed to examining metaphors in non-WEIRD populations in general in our future research. 

On p 15 lines 247-8, the authors write “After reading one of the three explanations associated with a particular question, participants were asked to consider how the general public would react to the message and to keep this in mind while responding to four questions.” I didn’t understand why the authors asked the participants to imagine how a general public audience would respond to the persuasive messaging. Isn’t the goal of the study to track how individual participants respond to the messaging? And to see how individuals respond to and are affected by the messages? It seems the participants should be reporting on their own impressions–how easy each participant found a passage to understand, how informative each participant found a passage–and not trying to speak for an imagined, general public audience.

***This is another helpful point from the reviewer. We chose to ask our participants to consider how the general public would react and to keep this in mind to mitigate any reluctance they might have to reveal their own views given the highly emotive and politicized nature of vaccine/booster discourse at the time of the study. As we know from decades of prior research, an audience (or halo) effect arises when a person’s behavior or reports about their own behavior or beliefs change because they believe someone else is watching them. Depersonalizing this and making it a less face-threatening task by asking participants to ostensibly report from a position of how the general public would react, rather than themselves, was a design factor intended to reduce: (a) the audience effect together with (b) any reluctance to go on the record with a personal response with the goal of obtaining more authentic data. We have now included some language to make the reasons behind this choice clear.

Not a point that needs to be addressed, but a possibly generative question I had while reading:

I was interested in how the results about which metaphors were effective and ineffective might reveal more about public understandings of public health more so than the effectiveness of individual metaphors. For example, the metaphors that rely on individualist ideas of health (i.e., do things that maximize your own personal health and minimize risks to your own persona health) were the more clear, salient, and effective metaphors (e.g., Bank metaphor, Castle metaphor, Raincoat metaphor) And the metaphors that asked audiences to think about health as an interdependent, community state, one in which we should make decisions toward a greater social good fell flat (the war metaphor). Do these findings suggest that specific metaphors themselves were more or less clear or do they suggest that metaphors, arguments, policies, etc that align with individualist notions of health make more sense than metaphors, arguments, polities, etc that align with collective approaches to health?

***We thank the reviewer for this very interesting point. We have added this possibility to the Limitations section in the General Discussion and included a reference to a recent study that contrasts the effects of information about personal vs. collective benefits of vaccinations during the Covid-19 pandemic (p. 37):

“Future work could also usefully consider differences in responses to metaphor depending on whether they emphasize the benefits of vaccination for the individual (e.g., Castle, Bank, Raincoat and Seatbelt metaphors) vs. benefit for others (e.g., Speed Limits and War metaphors). For example, a randomized control trial on the effects on U.K. adults of different types of vaccine information strategies during the COVID-19 pandemic found that, for strongly vaccine hesitant participants, information about the personal benefit of vaccination reduces hesitancy to a greater extent than information about collective benefits [55].” 

Sentence-level points

Is there a reason to use the word “Interestingly” on p 3 line 53? And “presciently”? (Global vaccine hesitancy was already a big problem in 2019, so more accurate than prescient, I would say).

***We agree with this point and have edited these sentences accordingly (p. 3):

“This was not a surprise for some observers. The World Health Organization (WHO) had named ‘vaccine hesitancy’ one of the top threats to global health in 2019 [1], months before the pandemic would take hold.”

Typo on p 4 line 87

***Thank you, we have now corrected this typo. 

I had trouble following the last 2 analytical sentences of the section “1.2. Explanatory metaphors in vaccine discourse” (on p. 8, lines 150 -6)

***We apologize for the confusion. We have now edited these sentences for clarity (pp. 8–9):

“Describing the influenza virus as a “beast” is quite different from the elaborate explanatory metaphors used in COVID-19 discourse, however. For one thing, “beast” is a metaphor for a virus, rather than a metaphor for some aspect of vaccination. For another, “beast” is a subtle, one-off metaphor rather than an extended and elaborated explanatory metaphor. In contrast, the “Cake” metaphor described above is explicitly presented as an explanation for how researchers could develop the COVID-19 vaccines so quickly, and it was extended and developed throughout the text. It is this latter type of explanatory metaphor that we aimed to evaluate in the present study.” 

6. PLOS authors have the option to publish the peer review history of their article (what does this mean?). If published, this will include your full peer review and any attached files.

Do you want your identity to be public for this peer review? For information about this choice, including consent withdrawal, please see our Privacy Policy.

Reviewer #1: No

Reviewer #2: Yes: Kari Campeau

---

## [Decision Letter · Decision Letter 1]

17 Aug 2023

PONE-D-23-09747R1Seatbelts and raincoats, or banks and castles: Investigating the impact of vaccine metaphorsPLOS ONE

Dear Dr. Flusberg,

Thank you for submitting your manuscript to PLOS ONE. After careful consideration, we feel that it has merit but does not fully meet PLOS ONE’s publication criteria as it currently stands. Therefore, we invite you to submit a revised version of the manuscript that addresses the points raised during the review process.

Thank you for improving the manuscript. In this round, two additional reviewers provided minor comments that should be fixed. On top of that, my comments are: (1) please use the singular form “between-subject” instead of “between-subjects,” (2) please provide p-values or at least the indication of the statistical significance of the differences in Table 4 (that was also asked by one of the Reviewers in the previous round), (3) when mentioning one-way ANOVA, please explicitly state what the factor (independent variable) is, (4) why you do not provide comparisons between two conditions that constitute your experimental design (i.e., metaphor vs. literal), as you declared (rows 188-189), (5) when reporting p-values on p. 20, please also report the means and test statistics.

We look forward to receiving your revised manuscript.

Kind regards,

Wojciech Trzebiński, Ph.D.

Academic Editor

PLOS ONE

Journal Requirements:

Reviewers' comments:

Reviewer's Responses to Questions

**Comments to the Author**

1. If the authors have adequately addressed your comments raised in a previous round of review and you feel that this manuscript is now acceptable for publication, you may indicate that here to bypass the “Comments to the Author” section, enter your conflict of interest statement in the “Confidential to Editor” section, and submit your "Accept" recommendation.

Reviewer #1: All comments have been addressed

Reviewer #3: All comments have been addressed

Reviewer #4: (No Response)

2. Is the manuscript technically sound, and do the data support the conclusions?

Reviewer #1: Yes

Reviewer #3: Yes

Reviewer #4: Yes

3. Has the statistical analysis been performed appropriately and rigorously? 

Reviewer #1: Yes

Reviewer #3: Yes

Reviewer #4: Yes

4. Have the authors made all data underlying the findings in their manuscript fully available?

Reviewer #1: Yes

Reviewer #3: Yes

Reviewer #4: Yes

5. Is the manuscript presented in an intelligible fashion and written in standard English?

Reviewer #1: Yes

Reviewer #3: Yes

Reviewer #4: Yes

6. Review Comments to the Author

Reviewer #1: (No Response)

Reviewer #3: As a "second-round" reviewer I would like to acknowledge that the authors answered to the concerns and question raised by previous reviewers and add that: 1) (lines 156-157) literature on the impact of metaphorical framing on vaccine attitudes is not limited to Scherer and colleagues 2015 (see Ervas et al. 2022); 2) (question 4; metaphor 2 WAR) "With certain viruses, some people who get infected can experience"  the word "can" is added, when compared to the other texts, with no apparent reason. In general, the texts (especially in the "literal version") are not well-balanced: the literal version is always shorter than the other versions and this might influence the ease and times for comprehension.

Reviewer #4: This paper describes a very interesting study with a sound methodology and relevant findings for the examined field (which also provide useful insights into aspects of metaphor use that go beyond the main purpose of the study itself). For instance, some inputs are given as to which specific aspects of vaccination can be successfully explained through metaphor use (e.g., natural immunity vs vaccine immunity). Furthermore, the section with the free-text answers (with some metaphors used incorrectly but with the same conceptual mapping presented in the explanation stimulus, others used correctly but in the attempt to refute the statements presented in the explanation stimulus, others invented by respondents) is also an interesting starting point for future research.

The study limitations (e.g., resorting to only a selection of metaphors available in vaccine campaigns) and previous reviews’ comments (like some remarks concerning methodology, like why respondents needed to state how they thought a general audience would react to certain messages rather than express their own opinion directly – to avoid some sort of “self-censorship” due to the stigmatization of vaccine aversion and hesitancy in an emotive and politicized historical moment like the one right after the coronavirus pandemic) seem to have been addressed properly.

The conclusion is that it is deemed to be highly advisable to publish this paper.

Some (minor) remarks and/or suggestions that may help further improve the paper are the following:

ABSTRACT:

I would mention the denomination “literal responses” a line before. The current formulation may be initially confusing: while reading the abstract, my first thought was that the study was going to examine extended metaphors and non-extended ones (as defined later in the paper - in p. 8, line 167 - “one-off” metaphors). A possible solution would be: “We created three response passages for each question: two included extended explanatory metaphors and one contained a literal response, with no explanatory metaphors”).

MAIN TEXT:

1) In the theoretical framework, when I read the section about extended explanatory metaphors (and their opposition to “one-off” metaphors) I thought about the frequent term “metaphor(ical) scenario” (as in Musolff 2006 or 2016*) with the “mininarratives”. It could be useful to add this terminology in the beginning to clarify what is meant in the text with “extended metaphors”. In case there is no full equivalence between this term and term “extended metaphors” as intended in the study, this could also be explained and motivated.

In addition, the term “metaphorical scenario” does indeed occur randomly in the paper (line 559, with respect to the Video Game metaphor, and line 574, with respect to the Raincoat metaphor), which further convinced me it could be useful to add it in the beginning, too.

N.B.: “scenario” also occurs a third time in line 583, in the segment “in the answers to the friend scenario”. As “scenario” is a technical term in metaphor studies, I would reformulate this part to avoid possible confusion.

2) I find that there are some similarities between some of the metaphors chosen for the study. More specifically, I feel that the Castle metaphor is a subtype of War metaphors (of course, with a different focus, but the mention of words like “invaders”, “defence” etc. seems to support this theory). I find this aspect important for several reasons. Firstly, on the basis of what is stated in the Linguistic Analysis at page 27: “[…] This is consistent with the finding that the War metaphor was perceived as particularly ineffective, as also shown in the ratings data described earlier. This is a particularly interesting finding given the prevalence of war messaging in vaccine discourse, particularly early on in the pandemic”. If the (more effective, “successful”) Castle metaphor is also regarded as a subtype of War metaphors, this once again shows that (although controversial!) war metaphors are quite effective, and this sentence should be modified accordingly.

To differentiate between these two metaphors, maybe a different (more specific, as occurs with Castle) label could be proposed for what is now called War metaphor in the paper survey (in Question 4, Metaphor 2), like Army Enrolment, Army Enlistment, or Mobilization.

Considering the Castle and Army Enrolment metaphors as subtypes belonging to the same source domain may also have an impact on the paper in other ways: as stated at page 24, some metaphors may be especially effective to communicate about some particular aspects of a phenomenon (e.g., Pilots and Fire Drills and a particular issue of natural immunity). If the categorization I put forward is implemented, the importance of choosing the right connection between a specific source and target (sub)domain is also better highlighted. So maybe war metaphors are effective in showing how vaccines work but not as effective in explaining the concept of herd immunity. This different effectiveness can also be explained through one of the variables reported in Table 1 at page 5 (army mobilization, unlike castle defence systems, is not very straightforward to many people! I also found this extended explanatory metaphor quite hard to follow as I first read it).

3) I think that the use of the rhetorical question “Would you rather wear a raincoat or walk around unprotected during a thunderstorm?” at page 31 should be highlighted, as it shows some sort of emotional involvement on part of the participant, which proves that this metaphor is effective.

TYPOS:

p. 19, line 326: five response questions (I guess!)

p. 20, line 359: explicit

p. 25, line 452: abstract

p. 29, line 29: “[…] question asked was “How do vaccines work?”. The […]

p. 31, line 577: “raincoat even if” (or, if it was a mistake made by the participant, add [sic]!)

p. 32, line 597: missing full stop before “finally”

p. 32, line 604: (92106; 100% re-user)  I think this parenthesis was not meant to be here, maybe it was a note/reference meant only for the authors which “slipped in” during writing. If this is not the case, it is not clear what this number refers to.

p. 36, line 688: there seems to be a missing space before “relatedly”

p. 37, line 711: after both occurrences of e.g. the comma is underlined

SOME POTENTIAL TYPOS/POINTS WITH POTENTIAL STYLISTIC IMPROVEMENTS:

Consider that I am not a native speaker of English, so these are some points that can be ignored by the authors if necessary:

p. 4, line 74: The seminal work […]?

p. 31, line 591: metaphor use? Or used?

p. 36, line 697: willingness/ the willingness to get vaccinated?

OTHER TYPOGRAPHICAL INDICATIONS:

I recommend being more typographically consistent in the way metaphors are reported. For instance, in the theoretical framework some metaphors are written in italics (beast, riot etc., line 158). In line 697, the same metaphors have a standard font, with no italics. In the introduction, some metaphors are reported between inverted commas (e.g., the “Cake” metaphor, line 168), but then they have a standard font in other parts of the text (e.g., the Bank or Castle metaphor, line 362). Of course, small capitals can be kept as a way to refer to conceptualizations and domains (as occurs in the introduction at page 4, for instance).

Line 662: seven in digits, maybe?

Use of hyphens: in some cases (like page 8, line 162-3, or 184-5) hyphens are longer and words are not separated by a space. In other cases, like at page 34 (line 658) the hyphen is shorter and there is a space that separates it from the words.

Pag. 18, line 313: maybe here the usual indication “[…]” should be used instead of the three dots?

*Musolff A. (2006) “Metaphor scenarios in public discourse”, Metaphor and Symbol, 21:1, pp. 23-38.

Musolff A. (2016) Political metaphor analysis: Discourse and scenarios, Bloomsbury Academic.

7. PLOS authors have the option to publish the peer review history of their article (what does this mean?). If published, this will include your full peer review and any attached files.

Reviewer #1: **Yes: **Amanda J Chase

Reviewer #3: No

Reviewer #4: No

---

## [Author Response · Author response to Decision Letter 1]

29 Sep 2023

All responses to the editor and reviewers is included in the "response to reviewers" document, which we submitted along with our revised manuscript.

---

## [Decision Letter · Decision Letter 2]

18 Oct 2023

PONE-D-23-09747R2Seatbelts and raincoats, or banks and castles: Investigating the impact of vaccine metaphorsPLOS ONE

Dear Dr. Flusberg,

Thank you for submitting your manuscript to PLOS ONE. After careful consideration, we feel that it has merit but does not fully meet PLOS ONE’s publication criteria as it currently stands. Therefore, we invite you to submit a revised version of the manuscript that addresses the points raised during the review process. Please submit your revised manuscript by Dec 02 2023 11:59PM. If you will need more time than this to complete your revisions, please reply to this message or contact the journal office at plosone@plos.org. Please include the following items when submitting your revised manuscript:A rebuttal letter that responds to each point raised by the academic editor and reviewer(s). You should upload this letter as a separate file labeled 'Response to Reviewers'.A marked-up copy of your manuscript that highlights changes made to the original version. You should upload this as a separate file labeled 'Revised Manuscript with Track Changes'.An unmarked version of your revised paper without tracked changes. You should upload this as a separate file labeled 'Manuscript'.If applicable, we recommend that you deposit your laboratory protocols in protocols.io to enhance the reproducibility of your results. Protocols.io assigns your protocol its own identifier (DOI) so that it can be cited independently in the future. For instructions see: https://journals.plos.org/plosone/s/submission-guidelines#loc-laboratory-protocols. Additionally, PLOS ONE offers an option for publishing peer-reviewed Lab Protocol articles, which describe protocols hosted on protocols.io. Read more information on sharing protocols at https://plos.org/protocols?utm_medium=editorial-email&utm_source=authorletters&utm_campaign=protocols.

We look forward to receiving your revised manuscript.

Kind regards,

Wojciech Trzebiński, Ph.D.

Academic Editor

PLOS ONE

Journal Requirements:

**Additional Editor Comments:**

 I appreciate your improvements, and the reviewers accept your revised manuscript. However, I have one more crucial doubt that needs to be addressed before the acceptance. Namely, one may understand from your Abstract that the only positive result you have consistently reached is that “participants in the metaphor condition provided longer free-response answers to the question posed by a hypothetical friend” (rows 39-41). If so, it is crucial to provide some argument that this difference is statistically significant. In Table 5, you report the “cumulative” word count. If it is a sum of words used by all participants in particular conditions, it depends on the sample sizes. So, it would probably be necessary to calculate the means of the word count per condition. Then, you should compare the means between conditions to check the statistical significance.

Please also double-check the manuscript for typos (e.g., row 566). 

Reviewers' comments:

Reviewer's Responses to Questions

**Comments to the Author**

1. If the authors have adequately addressed your comments raised in a previous round of review and you feel that this manuscript is now acceptable for publication, you may indicate that here to bypass the “Comments to the Author” section, enter your conflict of interest statement in the “Confidential to Editor” section, and submit your "Accept" recommendation.

Reviewer #3: All comments have been addressed

Reviewer #4: All comments have been addressed

2. Is the manuscript technically sound, and do the data support the conclusions?

Reviewer #3: Yes

Reviewer #4: Yes

3. Has the statistical analysis been performed appropriately and rigorously? 

Reviewer #3: Yes

Reviewer #4: Yes

4. Have the authors made all data underlying the findings in their manuscript fully available?

Reviewer #3: Yes

Reviewer #4: Yes

5. Is the manuscript presented in an intelligible fashion and written in standard English?

Reviewer #3: Yes

Reviewer #4: Yes

6. Review Comments to the Author

Reviewer #3: The authors addressed all my questions/concerns. Thanks for the opportunity to read this piece of reasearch.

Reviewer #4: At this stage of review, I only want to point out a few typos/stylistic suggestions:

War mobilization in italics, lines 526-7

line 632: and not capitalized

line 619-620: and not capitalized twice

We return to this finding (lines 470, 544): I think this could be formulated in a better way stylistically, like "This finding will be further discussed/discussed in greater detail ...etc. "

But see 37 (line 670): I find this could also be expressed in a better way

Posttest is sometimes written with an hyphen, sometimes without it; I do not understand if there is a reason to it

7. PLOS authors have the option to publish the peer review history of their article (what does this mean?). If published, this will include your full peer review and any attached files.

Reviewer #3: No

Reviewer #4: No

---

## [Author Response · Author response to Decision Letter 2]

3 Nov 2023

I appreciate your improvements, and the reviewers accept your revised manuscript. However, I have one more crucial doubt that needs to be addressed before the acceptance. Namely, one may understand from your Abstract that the only positive result you have consistently reached is that “participants in the metaphor condition provided longer free-response answers to the question posed by a hypothetical friend” (rows 39-41). If so, it is crucial to provide some argument that this difference is statistically significant. In Table 5, you report the “cumulative” word count. If it is a sum of words used by all participants in particular conditions, it depends on the sample sizes. So, it would probably be necessary to calculate the means of the word count per condition. Then, you should compare the means between conditions to check the statistical significance.

***We appreciate and agree with this suggestion. We have now conducted the appropriate calculations and statistical tests, which we have incorporated into the updated manuscript (pages 27-8). The inferential statistics provide further support for the conclusion presented in the abstract and general discussion: participants in the metaphor condition used significantly more words on average (~18% overall) compared to participants in the literal condition in response to the free text “explain to a friend” prompt across all the questions. 

***We have expanded Table 5 in this section of the paper to provide more descriptive data about the mean word counts for each question and condition. All our in-text discussion of these findings, as well as a revised Figure 3, now present the mean rather than cumulative word count data. We feel that presenting the data in this way and making these changes has strengthened this section of the paper as well as our overall conclusions. 

***Of note, while conducting these new analyses we discovered a single numerical error in our earlier manuscript. Specifically, we had under-reported the cumulative word count in the free text response for the war metaphor passage in response to Question 4. This appears to have been a random transcription error as we compiled the original manuscript, as an early version of the table we created included the correct number. This error is minor and only served to weaken our original conclusions. We have corrected the error and modified our discussion accordingly. We now have even stronger evidence to support our conclusion that participants who received a metaphorical passage used more words in their “explain to a friend” free text responses.

***Considering all this, however, we took this opportunity to double check all the analyses and values reported throughout the paper (in figures and text) to ensure accuracy. We discovered no other errors in the manuscript, so we are confident in the accuracy of our reporting. In the spirit of transparency, our raw data is accessible via an updated, now public link to the Open Science Framework. 

Please also double-check the manuscript for typos (e.g., row 566).

***We have now double-checked the manuscript for typos and other typographical errors and believe everything is in order. 

6. Review Comments to the Author

Reviewer #3: The authors addressed all my questions/concerns. Thanks for the opportunity to read this piece of research.

Reviewer #4: At this stage of review, I only want to point out a few typos/stylistic suggestions:

War mobilization in italics, lines 526-7

line 632: and not capitalized

line 619-620: and not capitalized twice

We return to this finding (lines 470, 544): I think this could be formulated in a better way stylistically, like "This finding will be further discussed/discussed in greater detail ...etc. "

But see 37 (line 670): I find this could also be expressed in a better way

Posttest is sometimes written with an hyphen, sometimes without it; I do not understand if there is a reason to it

***We have addressed all these issues in the revised manuscript.

---

## [Editor Report · Decision Letter 3]

8 Nov 2023

Seatbelts and raincoats, or banks and castles: Investigating the impact of vaccine metaphors

PONE-D-23-09747R3

Dear Dr. Flusberg,

We’re pleased to inform you that your manuscript has been judged scientifically suitable for publication and will be formally accepted for publication once it meets all outstanding technical requirements.

Kind regards,

Wojciech Trzebiński, Ph.D.

Academic Editor

PLOS ONE

Additional Editor Comments (optional):

Thank you for further improving your manuscript, especially in terms of statistical analysis. 

Minor issue to be fixed with the editorial team: at the oend of line 565, there is still an unnecessary quotation mark to be removed.
---

## [Editor Report · Acceptance letter]

13 Dec 2023

PONE-D-23-09747R3 

PLOS ONE

Dear Dr. Flusberg, 

I'm pleased to inform you that your manuscript has been deemed suitable for publication in PLOS ONE. Congratulations! Your manuscript is now being handed over to our production team.

Kind regards, 

on behalf of

Dr. Wojciech Trzebiński 

Academic Editor

PLOS ONE